# PANOWORLD-X: GENERATING EXPLORABLE PANORAMIC WORLDS VIA SPHERE-AWARE VIDEO DIFFUSION

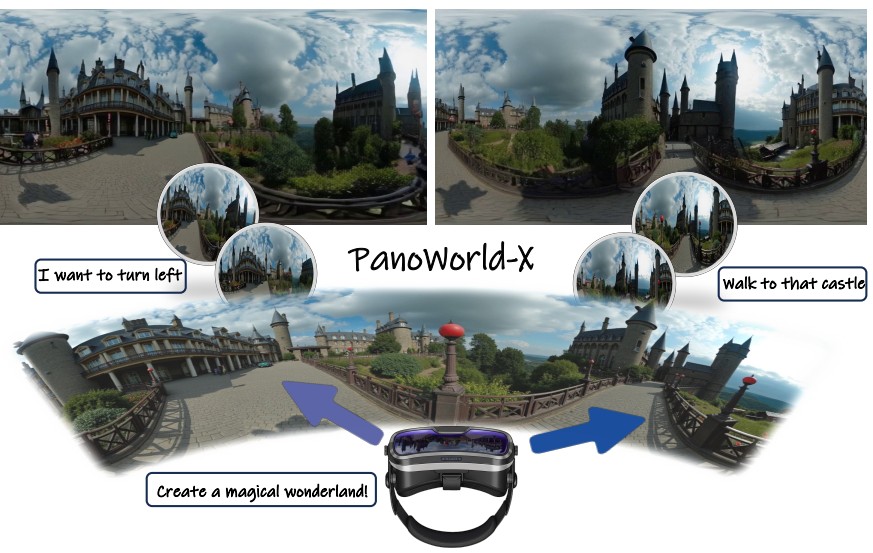

Figure 1: *PanoWorld-X* is a new framework for generating high-quality and controllable panoramic videos with diverse exploration paths and 360° visibility.

## ABSTRACT

Generating a complete and explorable 360-degree visual world enables a wide range of downstream applications. While prior works have advanced the field, they remain constrained by either narrow field-of-view limitations, which hinder the synthesis of continuous and holistic scenes, or insufficient controllability that restricts free exploration by users or autonomous agents. To address this, we propose *PanoWorld-X*, a novel framework for high-fidelity and controllable panoramic video generation with diverse camera trajectories. First, we propose a novel pipeline for synthesizing panoramic video-trajectory dataset pairs in virtual 3D environments via Unreal Engine. This pipeline consists of four main steps and enables the collection of a large-scale dataset with rich scene diversity and accurate trajectory annotations. To achieve precise panoramic video generation, we identify that the bottleneck arises from the misalignment between the spherical geometry of panoramic data and the inductive priors of conventional video diffusion models. To address this, we leverage the spherical connectivity characteristics of panorama data, and propose a Sphere-Aware Diffusion Transformer that reprojects equirectangular features onto the spherical surface, thereby capturing geometric adjacency in the latent space. This design significantly improves both visual fidelity and spatiotemporal continuity. Extensive experiments demonstrate that our PanoWorld-X achieves superior performance in various aspects, including motion range, control precision, and visual quality, underscoring its potential for real-world applications.

# 1 INTRODUCTION

The physical world is a 360-degree, fully explorable spatial environment where observers can perceive their surroundings from any angle. As an observer moves, the visual scene dynamically adapts to their changing position. Replicating these characteristics in the digital domain is critical to support diverse applications, such as the creation of immersive virtual reality (VR) environments (Yang et al., 2024a; Xanthidou et al., 2024) for human users and the development of simulated training spaces (Sora) for embodied intelligence (Liu et al., 2024c; Ma et al., 2024) and autonomous driving agents (Mao et al., 2023; Wang et al., 2024b). Thus, the generation of wide-field-of-view, explorable virtual worlds has emerged as a prominent research focus.

Towards this challenging objective, a technically feasible approach is to train large-scale video generation models (Yang et al., 2024c; Kong et al., 2024; Wang et al., 2025) using extensive real-world data. Taking Sora (Sora) as a prime example, it is considered a promising technology for simulating real-world environments. Nevertheless, these models usually can only generate perspective videos with a limited field of view, which restricts their capability to fully capture and represent entire scenes. Although certain methods attempt to generate long videos (Kim et al., 2024; Song et al., 2025) or 3D-aware videos (Liu et al., 2024a; Yu et al., 2024) with technical strategies to include more scene information, their outcomes still fall short in terms of geometric consistency and scene scale. Consequently, some researchers have shifted their focus to panoramic content generation. Thanks to its inherent characteristics, a world can be represented as a sequence of $360° \times 180°$ panorama images, which effectively captures scenes with consistency and a wide field of view.

Previous research on panorama generation has made substantial progress in creating static panoramic images (Zhang et al., 2024; Li & Bansal, 2023; Wu et al., 2023; Yang et al., 2024b; Feng et al., 2023; Ye et al., 2024). However, static images fail to address environmental incompleteness issues caused by occlusion. Panoramic video generation remains underexplored, with only a few studies (Wang et al., 2024a; Tan et al., 2024; Li et al., 2024; Liu et al., 2024b) tackling this challenge, yet several limitations persist: Firstly, **limited scenario diversity and restricted camera motion** in these works lead to poor performance in rendering occluded elements or distant small objects. Secondly, **lack of precise control over panoramic content** (e.g., specifying movement trajectories) limits interactivity with users or agents. Thirdly, **inadequate generation quality, characterized by spatiotemporal incoherence**, hinders generalization to new scenarios and real-world applications. We further identify that these limitations primarily arise from three key challenges: **(1) Scarcity of high-quality data**:Existing panorama datasets (Wang et al., 2024a; Tan et al., 2024) suffer from limited scale, scene diversity, and video motion dynamics. **(2) Limited fine control**: Although studies like (Ye et al., 2024) attempt to render training data, they still rely solely on textual input for generation, lacking finer-grained control mechanisms like trajectories. **(3) Neglect of panoramic geometry**: Existing methods treat panoramic data as perspective data, directly applying pre-trained perspective diffusion models without accounting for inherent geometric characteristics (e.g., spherical pixel distribution).

To address these challenges, we introduce **PanoWorld-X**, a novel framework for generating high-quality, explorable panoramic videos under movement signals control. We begin by creating a large, diverse dataset of panoramic videos with corresponding exploration paths. Our method introduces a new pipeline for preparing panoramic datasets using 3D scene rendering in Unreal Engine (UE). The process involves collecting multiple 3D scenes and implementing an automatic route sampling strategy to generate camera trajectories. We then filter out invalid paths through collision detection and render panoramic videos along the valid trajectories. We further employ the Video-LLaMA3 (Zhang et al., 2025) to filter low-quality samples and generate textual captions. As a result, we obtained 116,759 high-quality panoramic videos, each paired with its corresponding 3D exploration route, which we refer to as the **PanoExplorer dataset**.

To achieve explorable and high-quality panoramic video generation, we introduce Explorable Sphere-Aware DiT Block to replace the original DiT block of pre-trained video diffusion models. This block consists of three attention branches: the original global attention from the pre-trained diffusion model, an Exploration-Aware Attention introducing exploration route signal control, and a Sphere-Aware Attention to enhance spherical geometric perception and improve generation quality. The original global attention branch remains frozen, which enhances the efficiency and stability of training. Exploration-Aware Attention processes trajectories by first encoding them into specialized

embeddings based on their data characteristics. The embeddings are refined through attention layers and integrated via a zero-initialized projection layer for precise trajectory control.

Sphere-Aware Attention addresses the prior misalignment between panoramic and perspective data. In perspective images or videos generation, neighboring regions often contain related information, which leads pre-trained models to focus more on nearby tokens. However, panoramic data exhibit distinct characteristics. For example, the left/right edges and polar regions in the ERP panoramas are physically connected in 3D space but spatially separated in 2D projection images. Simply treating panorama data as perspective data when fine-tuning pre-trained models leads the models to underestimate their strong semantic correlations, resulting in reduced coherence. To address this issue, we adhere to the fundamental principle of panorama data by projecting ERP panoramas onto a spherical surface and calculating spatiotemporal spherical distances between patches. Sphere-Aware Attention then acts as a parallel branch that only attends to tokens within a spherical distance threshold, thus enhancing the neglected implicit correlations on the sphere. We find that this approach enhances generation quality and improves global consistency, particularly in the left-right boundary regions.

We conduct comprehensive experiments to evaluate the proposed framework. The results demonstrate that, compared to existing panoramic video generation models and camera controllable generation models, our PanoWorld-X exhibits superior performance in various aspects, including generation quality, motion range, and control precision. In summary, our key contribution includes:

- We propose a novel panorama dataset preparation pipeline in Unreal Engine and collect over 11K panoramic videos paired with trajectories. To the best of our knowledge, this is the largest synthetic panorama dataset featuring diverse scenes, extensive motion, and comprehensive text and trajectory annotations. We believe this dataset will provide significant value for future research.
- We introduce an Explorable Sphere-Aware DiT Block designed to bridge the gap between pre-trained video diffusion models and panorama data. This block consists of two key components: Exploration-Aware Attention for trajectory control and Sphere-Aware Attention, which leverages the spherical connectivity characteristics of panorama data to enhance the spatiotemporal consistency of generated outputs.
- Experimental results demonstrate that our approach outperforms previous methods and achieve large-scale movement, precise controllability, and high-quality panoramic video generation.

## 2 Related Work

Early video diffusion models Blattmann et al. (2023); Guo et al. (2023); Chen et al. (2023) extended image diffusion models Rombach et al. (2022) with temporal modules, but were limited by small model and dataset scales. Recently, Diffusion Transformers (DiTs) Peebles & Xie (2023) have enabled more scalable text-to-video models Yang et al. (2024c); Kong et al. (2024); Wang et al. (2025), greatly improving motion, temporal consistency, and video length. View-controllable video generation builds on this by introducing viewpoint changes. Some works Wang et al. (2024d); He et al. (2024); Liang et al. (2024); Bahmani et al. (2024); Liu et al. (2024a); Sun et al. (2024); Yu et al. (2024) use camera parameters to guide perspective, while others Valevski et al. (2024); Che et al. (2024) simulate device inputs for game-like interactivity. However, perspective-only data lacks full scene coverage, making it difficult to ensure 3D and cyclic consistency. Panorama image models Zhang et al. (2024); Li & Bansal (2023); Wu et al. (2023); Yang et al. (2024b); Feng et al. (2023); Ye et al. (2024) generate 360° images but cannot infer occluded regions. Panoramic video methods Wang et al. (2024a); Li et al. (2024); Liu et al. (2024b); Tan et al. (2024) expand spatial coverage but are restricted to small areas without substantial viewpoint progression. To address these limitations, we aim to generate explorable panoramic videos that combine large-scale spatial movement with 360° consistency, enabling the creation of coherent, interactive virtual worlds. More discussions about related works are in App. A

## 3 Method

To achieve high-quality panoramic video generation with free exploration, we first curate the PanoExplorer dataset in Section 3.2 and propose an Explorable Sphere-Aware DiT block in Section 3.3, which serves as a replacement for the original DiT block in pre-trained models.

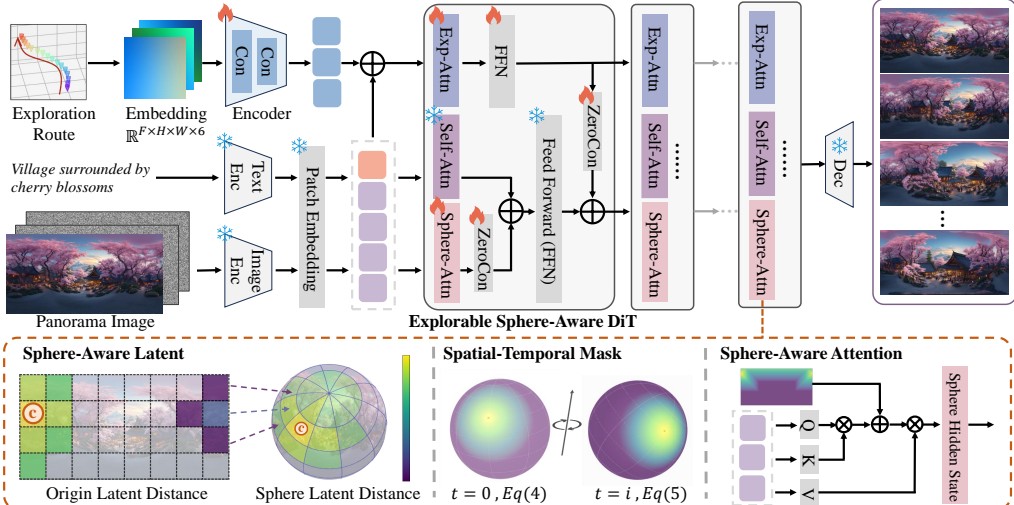

Figure 2: **PanoWorld-X Framework.** Given a panorama image with an exploration route, our model achieves high-quality and controllable panoramic video generation through the novel Explorable Sphere-Aware DiT blocks. This block employs the Exploration-Aware Attention to achieve precise view control, and the Sphere-Aware Attention to enhance spatiotemporal coherence by finding the related tokens through spherical geometry. For example, the upper-left token and the upper-right token are distant in the original latent space, but they are actually connected in reality. Therefore, we compute the distance on the sphere and use it as the basis for calculating the attention mask, which improves the interaction between associated tokens.

## 3.1 PRELIMINARY

**Video Diffusion Model.** The latest text-to-video generation models adopt latent-space multimodal DiT architectures, which consist of three key components: text encoders Radford et al. (2021); Raffel et al. (2020) for encoding textual prompts, a 3D-VAE Yu et al. (2023); Yang et al. (2024c) for video data compressing and tokenization, and a Transformer-based generator. As the core component, the generator is composed of numerous transformer blocks. It takes flattened video tokens and text tokens as inputs, modeling video and textual information through 3D self-attention (Yang et al., 2024c; Kong et al., 2024). However, this architecture presents certain challenges. On one hand, it incurs high computational and memory costs, making it more difficult to introduce additional control signals to guide the generation process. On the other hand, despite its global receptive field, the model sometimes overlooks essential long-range dependencies, resulting in decreased geometric consistency in the generated outputs. For panoramic video, which exhibits substantial distribution differences compared to common visual data, these issues become even more pronounced.

## 3.2 DATA CURATION

Collecting static panoramic videos with real-world exploration trajectories is challenging due to high labor costs and uncontrollable environments. To address this, we introduce the **PanoExplorer Dataset**, a scalable synthetic dataset for explorable Panoramic Video generation. We introduce a new dataset preparation pipeline built entirely in Unreal Engine. This pipeline ensures high-fidelity simulation, diverse environments, and precise trajectory control through four key construction stages.

*Step 1: Data Collection in Unreal Engine.* Unreal Engine enables flexible data preparation across diverse scenes, perspectives, and content. Using 504 high-fidelity 3D scenes, we cover varied indoor and outdoor environments with different weather and lighting, forming the basis for trajectory-based video recording.

*Step 2: Exploration Route Sampling.* We design a trajectory sampling algorithm to create plausible, visually coherent camera paths. For each scene, we first extract walkable surfaces (e.g., roads, floors) and apply Delaunay triangulation algorithm, which constructs a set of non-overlapping triangular meshes from sparse points on a two-dimensional plane. We then sample path candidates through

three steps: (1) randomly selecting two mesh points, (2) computing the shortest path with Dijkstra's algorithm, and (3) applying Laplacian smoothing to reduce abrupt turns. Only trajectories over 18 meters are retained to ensure sufficient temporal dynamics.

*Step 3: Collision Detection.* We introduce a collision detection mechanism to eliminate trajectories causing "geometry clipping" or "object intersections", which degrade quality and stability. We employ a bounding box proxy algorithm, in which objects are simplified to 3D bounding boxes based on their nearest and farthest points, balancing spatial accuracy and computational efficiency. Trajectories are simulated step-by-step, and any intersecting paths are discarded.

*Step 4: Data Annotation and Quality Filtering.* We ensure dataset quality through two filtering stages: (1) Automatic Filtering: We employ Video-LLaMA3 (Zhang et al., 2025) to assess videos based on detailed quality, semantic information, and motion richness, and subsequently filter out low-quality content. (2) Manual Assessment: The first frame of each video is manually screened to eliminate samples with poor rendering quality or missing details. Finally, Video-LLaMA3 automatically annotates videos to support text-controlled and multimodal tasks.

After this multi-step pipeline, we retain 116,759 high-quality static panoramic video sequences, each paired with its corresponding 3D exploration route. **For more detailed statistical analysis, please refer to the App B.** To our knowledge, this is **the largest synthetic panoramic dataset** available, featuring diverse scenes, extensive motion patterns, and comprehensive text and trajectory annotations. We hope that our data curation pipeline and scalable dataset pipeline will facilitate future research.

### 3.3 EXPLORABLE SPHERE-AWARE DiT BLOCK

In this section, we propose the Explorable Sphere-Aware DiT Block, replacing the original DiT block in pre-trained video diffusion models. This block effectively integrates viewpoint variation control signals of the exploration process with enhanced perception of spherical geometric features in panoramic data, enabling the generation of high-quality, explorable panoramic videos.

**Exploration Route Representation.** Previous methods for view-controllable video generation (He et al., 2024; Liang et al., 2024) typically use camera parameters such as intrinsic and extrinsic matrices to represent camera positions and viewpoints. However, our task faces two unique challenges. First, panoramic dataset does not contain intrinsic or extrinsic parameters, requiring us to design an equivalent transformation of the input signal. Second, our dataset consists of diverse 3D scene assets with vastly different absolute scales, using the original signal directly hinders model convergence.

To address these challenges, we design the exploration route representation. Movement commands are uniformly encoded as a six-degree-of-freedom control signal: $Er_i = (x_i, y_i, z_i, \alpha_i, \beta_i, \gamma_i)$ where $(x_i, y_i, z_i)$ denote spatial coordinates and $(\alpha_i, \beta_i, \gamma_i)$ represent yaw, pitch, and roll angles at position $i$. We first assume the center perspective of the panoramic image as the reference direction. Then, we convert the relative spatial coordinates $(\Delta x, \Delta y, \Delta z)$ (w.r.t. the first frame) into a translation matrix $T$ and transform the viewing angles $(\Delta \alpha, \Delta \beta, \Delta \gamma)$ into a rotation matrix $R$ using Euler angle principles, forming the extrinsic parameters. For the intrinsic parameters $K$, we make a simplified assumption based on game engine rendering rules—using a uniform focal length, fixing the principal point at the image center, and assuming no distortion. This transformation allows us to leverage the well-established Plücker embedding (Sitzmann et al., 2021) as a controllable representation, expressed as $(T \times d_{u,v}, d_{u,v})$. Here, $d_{u,v}$ denotes the direction vector of pixel $(u, v)$, computed as: $d_{u,v} = RK^{-1}[u, v, 1] + T$. We choose Plücker embedding because it provides pixel-level positional representation, enabling precise controllable generation—an approach widely adopted in prior work. (He et al., 2024; Liang et al., 2024). To address the impact of highly varying absolute scales, we introduce a new normalization strategy. First, we obtain metric depth values and select the 75th percentile as the anchor scale, which is set to 10 cm. Then, we rescale all spatial coordinates relative to this anchor frame. This strategy normalizes all scenes to a relative scale, ensuring that small scenes (e.g., indoor environments) do not exhibit overly exaggerated motion, while large scenes (e.g., outdoor natural landscapes) still maintain noticeable movement. Ablation studies confirm this normalization improves control fidelity and structural coherence

**Exploration-Aware Controllable Branch.** Pretrained diffusion transformer(DiT) models are trained on vast video datasets, making full fine-tuning computationally expensive. Therefore, we aim to

design an exploration route controllable branch that fulfills two critical goals: 1) enabling fine-tuning of large-scale diffusion transformer models with minimal data, and 2) maintaining the original latent space of the diffusion transformer to ensure output quality.

To achieve these objectives, we follow the principles of ControlNet (Zhang et al., 2023). We first encode the exploration route embedding using several 3D convolution layers to compress features into the same shape as the DiT latent. Then, we concatenate the video latent and condition latent to enable comprehensive information interaction. We initialize a new Exploration-Aware Attention (Exp-Attn) module with the same parameters as the original DiT block. To enhance training efficiency and minimize the impact on the original model, we employ a zero-linear layer, ensuring that the exploration route controllable branch initially has no effect on the original branch. Finally, we perform an element-wise addition of the outputs from the controllable branch and the original branch to achieve deep integration of information from both branches.

**Spherical Geometric Representation.** Previous studies (Wang et al., 2024a; Lu et al., 2024) fine-tune perspective generative models (Blattmann et al., 2023; Rombach et al., 2022) on panoramas, treating it as a process of adapting models to a specialized data domain. However, this straightforward fine-tuning strategy neglects the inherent properties of panorama data, as the pixels in panoramas are actually distributed in a sphere, which has significantly different geometric characteristics compared to plane geometry. Therefore, we introduce the following spatiotemporal distance representation based on spherical geometry, which has a crucial impact on the subsequent attention process.

In existing generative model frameworks, images or videos are encoded into a series of tokens. Previous 3D self-attention mechanisms enable the capture of spatially proximate tokens to gather more contextual information. Typically, different positional distances are determined based on planar geometry, where the distance between two points $p_1 = (x_1, y_1)$ and $p_2 = (x_2, y_2)$ is measured using the Euclidean distance:

$$d_{\text{Euclidean}}(p_1, p_2) = \sqrt{(x_2 - x_1)^2 + (y_2 - y_1)^2}. \tag{1}$$

While in panoramic ERP images, pixels distributed on a spherical surface introduce significantly different properties. For example, the point on the leftmost in Fig. 2 is physically connected with the points on the rightmost and those near the polar region in the spherical coordinates. However, when unfolded into a flattened image, these areas exhibit large spatial coordinate distances as in Eq. (1). Therefore, we introduce the spherical distance (Great-circle Distance) to accurately measure the distance between two points on the sphere. For an ERP image of width $W$ and height $H$, a pixel at coordinates $(x, y)$ can be converted to spherical coordinates $(\theta, \phi)$ as follows

$$\theta = \frac{2\pi x}{W} - \pi, \quad \phi = \frac{\pi y}{H} - \frac{\pi}{2}. \tag{2}$$

Here, $\theta$ represents the longitude (ranging from $-\pi$ to $\pi$), and $\phi$ represents the latitude (ranging from $-\frac{\pi}{2}$ to $\frac{\pi}{2}$). Building on geometric priors, we reproject the ERP panorama image onto a spherical surface and recalculate the distances. For two points with spherical coordinates $(\theta_1, \phi_1)$ and $(\theta_2, \phi_2)$, the spherical distance $d_{\text{spherical}}$ is given by the Haversine formula:

$$d_{\text{spherical}}(p_1, p_2) = 2R \cdot \arcsin\left(\sqrt{\sin^2\left(\frac{\phi_2 - \phi_1}{2}\right) + \cos(\phi_1)\cos(\phi_2)\sin^2\left(\frac{\theta_2 - \theta_1}{2}\right)}\right). \tag{3}$$

From a temporal perspective, movement in the real world can be interpreted as the rotation of the panoramic sphere in various directions. We define a rotation matrix $\mathbf{R}^{t_i}(\alpha, \beta, \gamma)$ that rotates a point on the sphere by Euler angles $\alpha$ (yaw), $\beta$ (pitch), and $\gamma$ (roll) at time $t_i$ relative to its initial orientation at time $t_0$. For a point $\mathbf{p}^{t_i} = (\theta, \phi)$ on the sphere, the rotated point $\hat{\mathbf{p}}^{t_i}$ relative to $t_0$ is given by

$$\hat{\mathbf{p}}^{t_i} = \mathbf{R}^{t_i}(\alpha, \beta, \gamma) \cdot \mathbf{p} = \mathbf{R}_z^{t_i}(\alpha) \cdot \mathbf{R}_y^{t_i}(\beta) \cdot \mathbf{R}_x^{t_i}(\gamma)\mathbf{p}, \tag{4}$$

where $\mathbf{R}_z(\alpha)$, $\mathbf{R}_y(\beta)$, and $\mathbf{R}_x(\gamma)$ are the rotation matrices around the z-axis, y-axis, and x-axis, respectively.So we measure the point $\mathbf{p}_1^{t_0}$ and $\mathbf{p}_2^{t_i}$ distance by first rotate to $\hat{\mathbf{p}}_2^{t_i}$ then calculate the sphere distance given in equation 3.

**Sphere-Aware Attention.** Based on the above analysis, we introduce Sphere-Aware Attention (Sphere-Attn) to enhance the generative model's perception of spherical geometric features. Specifically, we redefine the distances between latent representations based on their positions on the spherical

surface and attempt to enhance the response of closely located latent blocks. We set the attention mask to 1 for regions where the distance is below a specific threshold $\tau$, indicating areas that require mutual reinforcement. Mathematically, the spatial-temporal attention mask $M$ is defined as

$$M(p_1^{t_i}, p_2^{t_j}) = \begin{cases} 1 & \text{if } d_{\text{spherical}}(p_1^{t_i}, p_2^{t_j}) \leq \tau, \\ 0 & \text{otherwise.} \end{cases} \tag{5}$$

Then we replicate the components of the original 3D self-attention and convert video tokens to query $Q$, key $K$, and value $V$. Given the Sphere-Aware attention mask, our Sphere-Aware attention can be computed as

$$SphereAttn(Q, K, V, \mathbf{M}) = softmax(\frac{\mathbf{QK}^T}{\sqrt{d}} + \mathbf{M}) * \mathbf{V}. \tag{6}$$

As illustrated in Fig. 2, we employ a parallel mechanism. The video path simultaneously feeds into both the original self-attention block and the newly trained Sphere-Aware attention block. To ensure the Sphere-Aware attention block does not affect the initial training outcomes, a zero-initialized linear block is utilized. Compared to a sequential design, this parallel structure avoids significantly disrupting the output of each layer, which would otherwise lead to reduced training efficiency and increased difficulty in convergence.

## 4 EXPERIMENTATION

### 4.1 IMPLEMENTATION DETAIL

Since our model is based on a video diffusion framework, we select the advanced diffusion transformer model, CogVideoX-5B-I2V (Yang et al., 2024c). We fine-tuned the model to generate 49 frames with a resolution of $480 \times 720$. Due to the panoramic nature, the width-to-height ratio must adhere to 1:2. Therefore, we resized the output frames to $480 \times 960$. Compared to training a model with a native 1:2 aspect ratio, this post-processing resizing approach maximally preserves the prior knowledge of the original model. The model is trained on 8 A100 GPUs about 10,000 iterations. During the inference stage, we first generate a panorama image using a text prompt with FLUX (Labs, 2024), leveraging the panorama LoRA released by Yang (Yang et al., 2024b). Subsequently, we input the first image along with specific action signals into our model.

### 4.2 EVALUATION DATASETS AND METRICS

To evaluate the performance of explorable panoramic video generation, we randomly select 200 panoramic videos from our curated dataset. Each panoramic video includes a movement trajectory. We compare the generated videos with ground-truth video clips using multiple metrics:(1) Pixel-level visual quality is measured using PSNR, SSIM, and LPIPS. (2) Visual quality and temporal coherence are assessed using Frechet Inception Distance (FID) (Heusel et al., 2017) and Frechet Video Distance (FVD) (Unterthiner et al., 2019). (3) Exploration route control precision is evaluated using Rotation Error ($R_{err}$) and Translation Error ($T_{err}$) metrics, as introduced by He et al. (2024). These metrics compute the camera extrinsic parameters in comparison to the ground truth camera pose. To accommodate varying output lengths, the mean of these metrics is calculated rather than their sum.

### 4.3 QUANTITATIVE RESULTS

**Comparison with Panoramic Video Generation Models.** We evaluate the performance of our proposed methods by comparing them with state-of-the-art Panoramic Video generation models. Specifically, we select three baseline methods for comparison:(1) 360DVD (Wang et al., 2024a) uses a trainable 360-Adapter to extend standard T2V (Guo et al., 2023) models to the panorama domain. (2) Imagine360 (Tan et al., 2024) is a model designed to convert perspective videos into panoramic videos. We crop perspective videos from the ground truth videos and evaluate their generation capabilities. (3) GenEX (Lu et al., 2024) is a stable video diffusion (Blattmann et al., 2023) based model for Panoramic Video generation. Based on its official checkpoint, it only supports

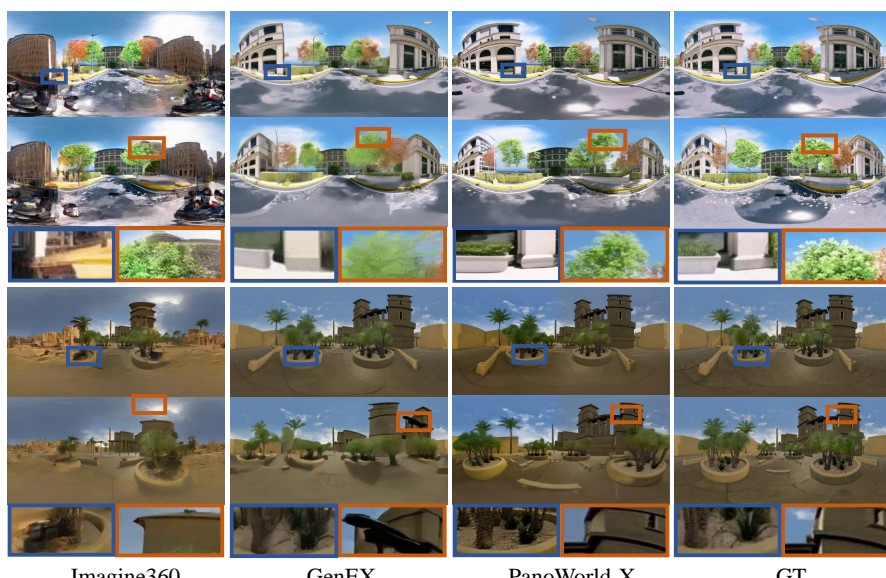

| Imagine360 | GenEX | PanoWorld-X | GT |

Figure 3: Qualitative comparisons with panoramic video generation models on generated keyframes. Our results exhibit superior detail clarity and geometric coherence throughout the movement. **Additional video visualization results are provided in the supplementary materials.**

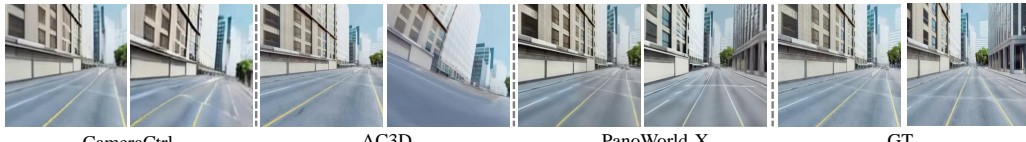

| CameraCtrl | AC3D | PanoWorld-X | GT |

Figure 4: Qualitative comparisons with camera controllable methods on cropped perspective views from panoramas. Our results demonstrate more precise camera control and generation quality.

image-to-video generation and lacks a controllable video generation checkpoint. Therefore, we only compare its video quality with our proposed methods.

The qualitative comparison shown in Fig. 6 demonstrates the high-quality generation capability of our model. Imagine360 struggles to consistently generate stable content on both sides, often leading to detail degradation. GenEX tends to produce blurry images during the generation process. In contrast, our method not only maintains excellent detail in the generated images but also achieves a significantly wider range of camera movement compared to other approaches. Additionally, Tab. 1 confirms that our method outperforms previous methods across all evaluation metrics.

**Comparison with Camera Controllable Generation Models.** To evaluate our explorable route-controllable Panoramic Video generation, there are no existing works with the same setting for direct comparison. As a result, we compare our approach with camera-controllable generation models. Since these models are not trained on panoramic datasets, we crop our results into perspective view videos and input the ground truth perspective view images into camera-controllable models to ensure a fair comparison. We select two state-of-the-art methods for this evaluation: CameraCtrl (He et al., 2024) and AC3D (Bahmani et al., 2024). We evaluate the generation results from two perspectives. First, in terms of image quality, as shown in Tab. 1, our method surpasses previous approaches. The quality of details is also evident in the comparison provided in Fig. 4. Second, regarding controllability, CameraCtrl often struggles to achieve precise control over content, typically allowing only very limited movement. While AC3D shows improvement in controllability compared to earlier methods, it still performs poorly when handling complex trajectories. In contrast, our method demonstrates a significant enhancement in controllability compared to previous approaches.

**Ablation Study.** We primarily investigate the importance of the proposed components from three aspects. **(1) Data Normalization:** As mentioned in Sec 3.3, we normalize the dataset stride scale, which enhances sensitivity to control signals and facilitates more significant camera movements.

Table 1: Comparison of Panoramic Video Generation and Camera Controllable Generation Models.

| Models | PSNR ↑ | SSIM ↑ | LPIPS ↓ | FID ↓ | FVD ↓ | $R_{err}$ ↓ | $T_{err}$ ↓ |
|---|---|---|---|---|---|---|---|
| *Panoramic Video Generation Models* | | | | | | | |
| 360DVD (Wang et al., 2024a) | 10.66 | 0.35 | 0.73 | 111.85 | 2049.21 | – | – |
| Imagine360 (Tan et al., 2024) | 11.62 | 0.39 | 0.591 | 66.73 | 1830.46 | – | – |
| GenEX (Lu et al., 2024) | 16.12 | 0.59 | 0.42 | 42.22 | 1113.72 | – | – |
| PanoWorld-X | **19.34** | **0.63** | **0.24** | **28.01** | **467.18** | – | – |
| *Camera Controllable Generation Models* | | | | | | | |
| CameraCtrl (He et al., 2024) | 11.56 | 0.38 | 0.61 | 108.12 | 2017.95 | 0.097 | 0.245 |
| AC3D (Bahmani et al., 2024) | 13.77 | 0.49 | 0.52 | 41.98 | 842.29 | 0.081 | 0.087 |
| PanoWorld-X Perspetive | **16.76** | **0.56** | **0.42** | **38.63** | **586.51** | **0.061** | **0.073** |

Table 2: Ablation Study on Individual Components.

| Models | PSNR ↑ | SSIM ↑ | LPIPS ↓ | FID ↓ | FVD ↓ | $R_{err}$ ↓ | $T_{err}$ ↓ |
|---|---|---|---|---|---|---|---|
| w/o Data Normalization | 17.11 | 0.55 | 0.32 | 40.37 | 751.18 | 0.114 | 0.102 |
| w/o Controllable Branch | 16.30 | 0.53 | 0.36 | 38.71 | 769.42 | 0.102 | 0.152 |
| w/o Sphere-Aware Attention | 17.59 | 0.56 | 0.27 | 29.96 | 492.98 | 0.069 | 0.076 |
| Full model | **19.34** | **0.63** | **0.24** | **28.01** | **467.18** | **0.061** | **0.073** |

**(2) Exploration Route Controllable Branch:** Previous works were unable to precisely control the direction of generated content movement. With this module, we can achieve forward movement in the generated panorama video while enabling rotation in any direction. Quantitatively, this approach significantly reduces both $R_{err}$ and $T_{err}$. **(3) Sphere-Aware Attention:** Panoramic videos should maintain left-right continuity, but achieving seamless stitching remains challenging. As shown in Fig. 5 (b), the model without sphere-aware attention exhibits visible vertical seams at the left-right boundaries, whereas the full model achieves harmonious results. The primary improvement stems from the sphere-aware attention mechanism, which enhances interhemispheric coherence by referencing the most relevant patches on the spherical surface. Tab 2 demonstrates significant improvements across all metrics and Fig. 5 (a) show detail quality improvement. More detailed ablation studies on the parameter settings for sphere-aware attention can be found in App. C.

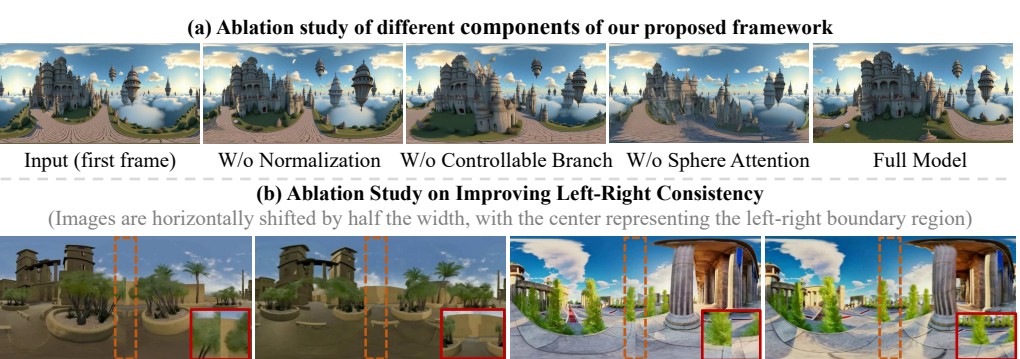

**(a) Ablation study of different components of our proposed framework**

Input (first frame)  W/o Normalization  W/o Controllable Branch  W/o Sphere Attention  Full Model

**(b) Ablation Study on Improving Left-Right Consistency**
(Images are horizontally shifted by half the width, with the center representing the left-right boundary region)

W/o Sphere-Aware Attn  Full Model  W/o Sphere-Aware Attn  Full Model

Figure 5: Ablation study. The data normalization strategy improves the range of motion. The controllable branch enhances control precision. The sphere-aware attention significantly enhances the quality of panoramic details, particularly in terms of left-right consistency.

## 5 CONCLUSION

We introduce PanoWorld-X, a novel framework designed for generating explorable panoramic videos. It addresses two key limitations of prior methods: the narrow field of view in traditional video generation models and the issues of uncontrollable camera movements and limited motion range in existing panorama generation approaches. Leveraging a curated dataset, we design a controllable

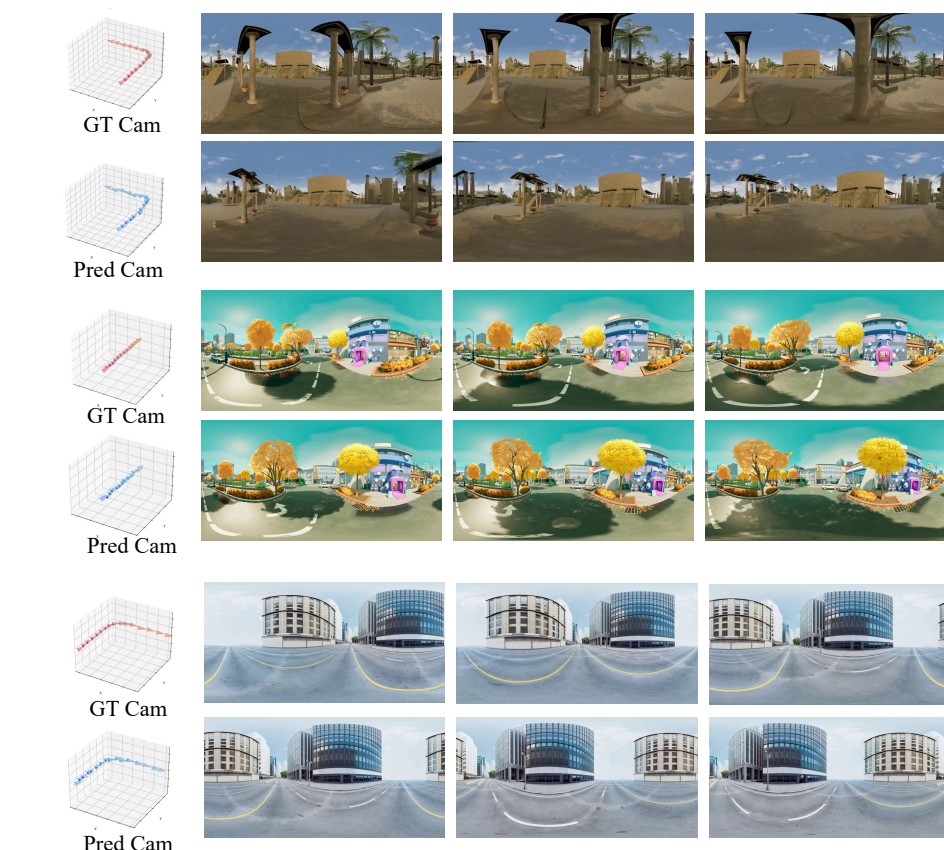

Figure 6: Additional visualization results for Reviewer 8Add. We present three new scenarios: move forward and turn left, move forward, and move forward and turn right. The predicted cameras are computed using VGGT. We compare the predicted camera trajectories with the ground truth as well as the corresponding image visualizations. Overall, the predictions are highly consistent with the ground truth camera trajectories, which is also aligned with the quantitative evaluation results.

branch to enable precise exploration route control and employ a sphere-aware attention mechanism to enhance visual quality. Our evaluations demonstrate that PanoWorld-X outperforms previous state-of-the-art methods. We believe our work will inspire future research in this domain.

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

# Supplementary Materials for
## *PanoWorld-X: Generating Explorable Panoramic Worlds via Sphere-Aware Video Diffusion*

CONTENTS

## A   MORE RELATED WORK

**Video Diffusion Models.** Early research on video diffusion models Blattmann et al. (2023); Guo et al. (2023); Chen et al. (2023) primarily focused on extending pre-trained image diffusion models Rombach et al. (2022) to video generation. These studies achieved dimensional extension by incorporating temporal interactions into the UNet architecture of image diffusion models. However, the limited scale of both models and datasets constrained the quality of the generated results. More recently, Diffusion Transformers (DiTs) Peebles & Xie (2023) were introduced, with authors demonstrating that Transformer-based Vaswani et al. (2017); Dosovitskiy et al. (2020) generators exhibit superior scalability compared to UNet-based architectures. Over the past year, several DiT-based text-to-video models Yang et al. (2024c); Kong et al. (2024); Wang et al. (2025) have emerged. Their successful scaling has significantly improved the performance in various aspects, including generated duration, motion amplitude, and temporal consistency, thus providing a more robust prior for a wide range of downstream tasks.

**View-Controllable Video Generation.** Generating video that matches a sequence of changed views is a critical step toward creating a virtual world and applying it to scenarios such as agent systems. Existing research can be broadly categorized into two main types. The first one Wang et al. (2024d); He et al. (2024); Liang et al. (2024); Bahmani et al. (2024); Liu et al. (2024a); Sun et al. (2024); Yu et al. (2024) focuses on incorporating camera extrinsic parameters as additional input signal into pretrained video generation models, allowing the generated video to have perspective changes according to the position and orientation of the camera. The second type Valevski et al. (2024); Che et al. (2024) aims to generate video games, achieving video-game-like interactivity. The viewpoint changes in these works are controlled by simulating inputs from devices such as keyboards. However, these methods face significant limitations due to the constrained information provided by perspective-view videos. They struggle to generate complete scenes while maintaining 3D consistency and cyclic consistency, leading to notable instability in the generated scenes.

**Panorama Generation Models.** Benefiting from the rapid advancements in 2D image generation, diffusion-based panorama image generation models Zhang et al. (2024); Li & Bansal (2023); Wu et al. (2023); Yang et al. (2024b); Feng et al. (2023); Ye et al. (2024) has achieved significant results. Despite the 360-degree visible nature of panoramas, physically occluded content (e.g., what lies around the next corner) remains difficult to obtain. Consequently, Panoramic video is necessary to capture broader spatial information, facilitating the construction of a comprehensive world model. 360DVD Wang et al. (2024a) initially established the WEB360 panoramic video dataset and accomplished text-to-Panoramic Video generation. Methods such as 4K4DGen Li et al. (2024), DynamicScaler Liu et al. (2024b), and Imagine360 Tan et al. (2024) achieved dynamic panoramic video generation with object movements. However, these approaches are unable to generate content with significant viewpoint progression, restricting the scope of world content generation to relatively limited areas. Furthermore, these methods fail to achieve precise and controllable 360-degree worlds, which hinders the ability to interact effectively with the generated worlds. The most related work is GenEX Lu et al. (2024). However, it does not incorporate the geometric properties of panoramas and instead simply fine-tunes the model, resulting in insufficient quality in the final details. Therefore, our objective is to generate explorable panoramic videos that incorporate extensive spatial movement.

## B   DETAILS OF THE CURATED DATASET

### B.1   VISUALIZATION OF DATA PAIRS

This subsection provides an overview of the curated dataset. Our PanoExplorer Dataset comprises 116,759 high-quality data pairs. Each pair consists of a panoramic video sequence and its corresponding exploration route. We present several samples from the dataset to provide an intuitive understanding of its structure and stylistic characteristics in Fig. A.

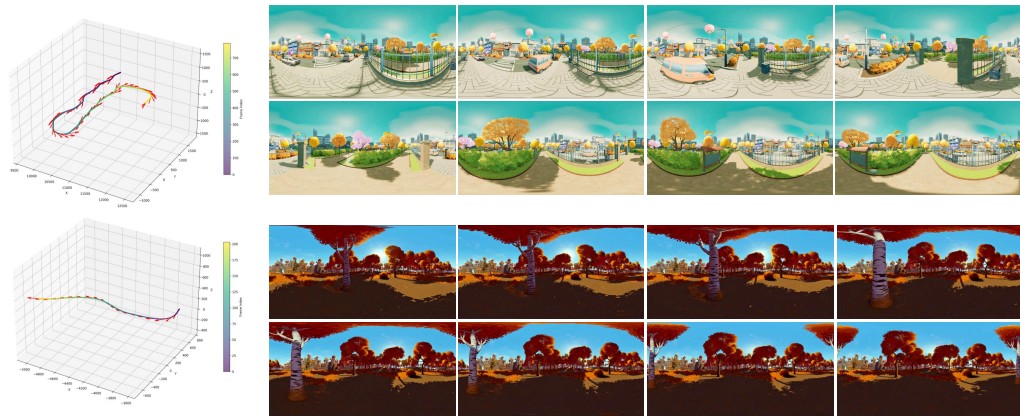

Figure A: **PanoExplorer Datasets.** We present two samples from the datasets. The sequences, arranged from left to right and top to bottom, represent video frames evolving along the timeline. The left side illustrates the visualized exploration routes.

## B.2 STATISTICAL ANALYSIS OF SCENE CATEGORIES

To demonstrate the diversity of our dataset, we provide a more comprehensive statistical analysis. Using the captions and first images as inputs, we employ GPT-4o to classify scene types. The statistical results demonstrate that our dataset covers most scene categories, exhibiting substantial diversity.

Table A: Scene categories with descriptions and percentages.

| Category | Description | Percentage |
|---|---|---|
| Natural Landscape | Mountains, forests, oceans, deserts, etc. | 16.86% |
| Urban Street Scene | Streets, plazas, building exteriors, sidewalks, etc. | 13.26% |
| Indoor Space | Homes, offices, stores, classrooms, etc. | 19.38% |
| Transportation Scene | Airports, stations, roads, railways, bridges, etc. | 9.64% |
| Agricultural/Rural Scene | Fields, farms, villages, orchards, etc. | 8.72% |
| Water-Based Scene | Rivers, lakes, dams, harbors, coasts, etc. | 6.52% |
| Disaster/Extreme Weather | Floods, fires, earthquakes, snowstorms, etc. | 5.14% |
| Sports Scene | Stadiums, arenas, sports fields, training areas, etc. | 7.68% |
| Religious/Ceremonial Scene | Temples, churches, weddings, funerals, rituals, etc. | 4.21% |
| Virtual/Science Fiction | Game environments, futuristic cities, virtual worlds, etc. | 8.59% |

## B.3 ANALYSIS OF TRAJECTORY COMPLEXITY

To further analyze camera motion patterns, we quantify trajectory complexity using tortuosity, following Benhamou (2004). Tortuosity(T) is defined as the ratio of the straight-line distance(D) between the start and end points to the actual path length(L), formulated as T=D/L. This metric ranges between 0 and 1, where: A value closer to 1 indicates a near-straight path. A lower value reflects a more complex, winding trajectory. We calculate all trajectories and give the statistics as followed.

The statistical results indicate that the trajectories in our dataset exhibit diversity and follow a long-tailed distribution.

## B.4 PROCESS OF GENERATING TEXT CAPTIONS WITH EXAMPLES

Our caption is generated by Video-LLaMA3 Zhang et al. (2025), which is the state-of-the-art VLM model for video caption. We give Video-LLaMA3 the prompt like 'Generate a detailed, temporally-aware caption for this panoramic video. Describe the environment (indoor/outdoor, natural/urban, time of day, weather, lighting......), key subjects (objects/people/animals with appearances, actions,

Table B: Tortuosity ranges with corresponding percentages and interpretations.

| Tortuosity Range | Percentage | Meaning Interpretation |
|---|---|---|
| 1 | 7.13% | Perfectly straight path |
| $1 > T \geq 0.75$ | 60.24% | Noticeable directional changes, possibly with one or two turns |
| $0.75 > T \geq 0.5$ | 32.29% | Significant directional changes, possibly with multiple turns |
| $0.5 > T \geq 0$ | 0.34% | Highly complex trajectories |

and interactions......), and camera dynamics (static/smooth rotation/fast panning, direction, perspective, and horizon tilt......). Note temporal changes (new subjects entering, lighting shifts......) and spatial details (foreground/background layers......).'

An example of caption: "The video showcases a vibrant, alien landscape with tall, mushroom-like trees and glowing plants. The scene is bathed in a deep blue light, creating an otherworldly atmosphere. The camera slowly pans to the right, revealing more of the fantastical environment. The sky is filled with stars, and a large, glowing moon hangs in the distance. The ground is covered in lush, blue grass, and there are large, rocky formations scattered throughout the landscape. The overall effect is one of wonder and exploration." This example demonstrates that our captions provide comprehensive descriptions (approximately 100 words) while effectively conveying multiple aspects of the scene.

### B.5 COMPARISON WITH OTHER PANORAMA DATASETS

We compare our *PanoExplorer* dataset with existing panoramic datasets. Web360 Wang et al. (2024a) is one of the early panoramic generation datasets, sourced from 360-degree camera videos on YouTube. Imagine360 Tan et al. (2024) and PanoWan Xia et al. (2025) expanded on this by increasing the volume and diversity of the data. However, these datasets primarily focus on outdoor natural landscapes, which inherently limit the range of motion in the scenes and lack trajectory information. 360-1M Wallingford et al. (2024) is currently one of the larger panoramic datasets, also filtered from 360-degree videos on YouTube. However, it lacks textual annotations and trajectory data, requiring reliance on additional models like Colmap or Dust3R Wang et al. (2024c) for reconstruction. This introduces potential inaccuracies, making it unsuitable for tasks requiring precise trajectory control. DiffPano Ye et al. (2024) utilizes the Habitat Simulator to render a panoramic video dataset based on the Habitat Matterport 3D (HM3D) dataset Ramakrishnan et al. (2021). Similar to our approach, DiffPano employs synthetic data generation. However, it is limited to indoor scenes, with motion restricted to movement within and between rooms, and the dataset size remains relatively small.

Compared to previous datasets in Tab. C, our *PanoExplorer* dataset offers several advantages:

1. **Richer Scene Diversity** – As detailed in Sec. B.2, our method enables the collection of diverse panoramic videos by leveraging various 3D assets.

2. **More Controllable Trajectories** – Real-world captured videos lack trajectory information, requiring estimation through additional models, which can introduce errors. In contrast, virtual environments allow precise trajectory control and easy storage of motion data, ensuring accurate paired samples.

3. **Larger-Scale Dataset Collection** – Unlike previous approaches that rely on costly real-world data collection and manual annotation, our method generates a significantly larger dataset at a fraction of the cost.

## C ADDITIONAL ABLATION STUDIES AND ANALYSES

To further analyze our proposed module, we conduct additional ablation studies and analyses focusing on the sphere attention mechanism.

Table C: Comparison of panoramic datasets.

| Dataset | Samples | Indoor | Outdoor | Diversity | Motion | Trajectory | Text | Origin |
|---|---|---|---|---|---|---|---|---|
| Web360 | 2,114 | × | ✓ | Low | Low | × | ✓ | Real-World |
| Imagine360 | 10,744 | × | ✓ | Medium | Medium | × | ✓ | Real-World |
| PanoWan | 13,000 | ✓ | ✓ | Medium | Medium | × | ✓ | Real-World |
| 360-1M | 1,076,592 | ✓ | ✓ | High | High | × | × | Real-World |
| DiffPano | 8,508 | ✓ | × | Low | Low | × | ✓ | Synthetic |
| Ours | **116,759** | ✓ | ✓ | **High** | **High** | ✓ | ✓ | Synthetic |

Table D: Training convergence speed and performance comparison with and without Sphere-Aware Attention.

| Model | PSNR | Avg Loss |
|---|---|---|
| w/o Sphere-Aware Attention 4000 step | 14.02 | 0.1282 |
| w/o Sphere-Aware Attention 7000 step | 15.36 | 0.0942 |
| w/o Sphere-Aware Attention 10000 step | 17.59 | 0.0794 |
| with Sphere-Aware Attention 4000 step | 16.74 | 0.1082 |
| with Sphere-Aware Attention 7000 step | 19.08 | 0.0687 |
| with Sphere-Aware Attention 10000 step | **19.34** | **0.0639** |

### C.1 IMPACT OF FASTER CONVERGENCE

Our sphere-aware attention design adheres to the principles of panoramic sphere projection, thereby providing additional prior information to guide the model's convergence direction. As shown in Tab. D We compared the results across different training epochs with and without sphere-aware attention. The findings indicate that models incorporating sphere-aware attention achieve convergence earlier than those without it. Specifically, with sphere-aware attention, the model significantly outperforms the without sphere-aware attention by 7000 steps, achieving results comparable to those of the baseline at 10,000 steps. Furthermore, it demonstrates a clear trend toward convergence at this earlier stage. The experimental results further validate our theoretical analysis.

### C.2 IMPACT OF SPHERE-AWARE MASK THRESHOLD

As the mask threshold increases, the sphere-aware attention mechanism increasingly resembles standard 3D self-attention. In the limiting case where threshold = 1 (activating all regions), the formulation reduces exactly to pure self-attention. Conversely, decreasing the threshold narrows the receptive field, effectively limiting the model's perception to closer regions on the spherical representation. We analyze this parameter variation in the following ablation study, as shown in Tab. E.

### C.3 IMPACT OF DYNAMIC TEMPORAL MASK

Our sphere-aware mask incorporates both spatial (inter-frame) and temporal (cross-frame) components. The spatial mask captures neighborhood region information, while the temporal mask models relationships between corresponding regions across frames. Since pixel positions shift over time due

Table E: Comparison of mask threshold variants of our model.

| Model | PSNR | SSIM | LPIPS | FID | FVD | $R_{err}$ | $T_{err}$ |
|---|---|---|---|---|---|---|---|
| Baseline (w/o Sphere-Aware Attention) | 17.59 | 0.56 | 0.27 | 29.96 | 492.98 | 0.069 | 0.076 |
| Self-Attention (Threshold=1) | 17.94 | 0.58 | 0.29 | 29.03 | 489.03 | 0.065 | 0.077 |
| Threshold=0.25 | 18.68 | 0.61 | 0.25 | 28.77 | 473.01 | 0.062 | 0.075 |
| Full Model (Threshold=0.5) | **19.34** | **0.63** | **0.24** | **28.01** | **467.18** | **0.061** | **0.073** |

Table F: Ablation study of dynamic temporal mask.

| Model | PSNR | SSIM | LPIPS | FID | FVD | $R_{err}$ | $T_{err}$ |
|---|---|---|---|---|---|---|---|
| Baseline (w/o Sphere-Aware Attention) | 17.59 | 0.56 | 0.27 | 29.96 | 492.98 | 0.069 | 0.076 |
| w/o dynamic temporal mask (remove Eq.(5)) | 18.26 | 0.60 | 0.26 | 28.87 | 472.45 | 0.063 | 0.075 |
| Full Model (spatial+temporal) | **19.34** | **0.63** | **0.24** | **28.01** | **467.18** | **0.061** | **0.073** |

to viewpoint changes, we incorporate the transformation in Eq. 5. This ensures that the temporal (cross-frame) attention masks dynamically adjust according to the relative rotation between frames, maintaining geometric consistency in the spherical domain.

To analyze this design choice, we conducted an ablation study by relaxing our assumption: when removing the transformation in Eq. 5 and ignoring viewpoint-induced pixel displacement, the model only considers adjacent regions within frames and identical pixel regions across frames. The experiment results are in Tab. F.

These results demonstrate that simplifying the temporal mask leads to performance degradation, yet still yields improvement over the baseline. This indicates that for models incorporating spherical geometric priors, more accurate geometric guidance provides greater benefits during training.

# D    MORE VISUALIZATION OF PANORAMIC VIDEO GENERATION

## D.1    EVALUATION OF ZERO-SHOT GENERALIZATION ON IN-THE-WILD IMAGES

To verify our model's zero-shot generalization capability, we use diverse in-the-wild images to generate panoramic videos. The first images are either pre-existing or generated. As illustrated in Fig. B, we evaluate our model using images from diverse environments, including outdoor, indoor, real-world scenes, and synthetic scenarios. The generated results demonstrate high-quality details and accurate geometric representation. The results demonstrate the ability to handle various exploration routes, such as moving forward, turning left, and turning right. All visualization results are available in **Video1-In-the-wild-results.mp4**.

## D.2    COMPARISON ON THE TEST SET

In this section, we compare our method with state-of-the-art panoramic video generation methods on test sets to evaluate the quality of the generated outputs. As illustrated in Fig. C, our results exhibit clearer details and better image quality. All visualization results are available in **Video2-Pan-Comparison.mp4**.

To further evaluate the quality of perspective videos compared to panoramic video generation methods, we divide the panoramic video into four main sections: front, left, right, and back. Specifically, we convert the panoramic image into four perspective images. As shown in Tab. G, our method outperforms previous state-of-the-art approaches in terms of generation quality across all directions. This demonstrates that our results not only achieve better global generation quality but also surpass previous methods in fine local details.

Table G: Comparison of PSNR on different viewing directions with orther method.

| Method | Front | Left | Right | Back |
|---|---|---|---|---|
| 360DVD | 8.94 | 9.37 | 9.11 | 9.25 |
| Imagine360 | 16.06 | 10.97 | 10.95 | 9.64 |
| GenEX | 15.89 | 15.77 | 15.99 | 16.17 |
| Ours | **16.76** | **16.67** | **17.04** | **17.31** |

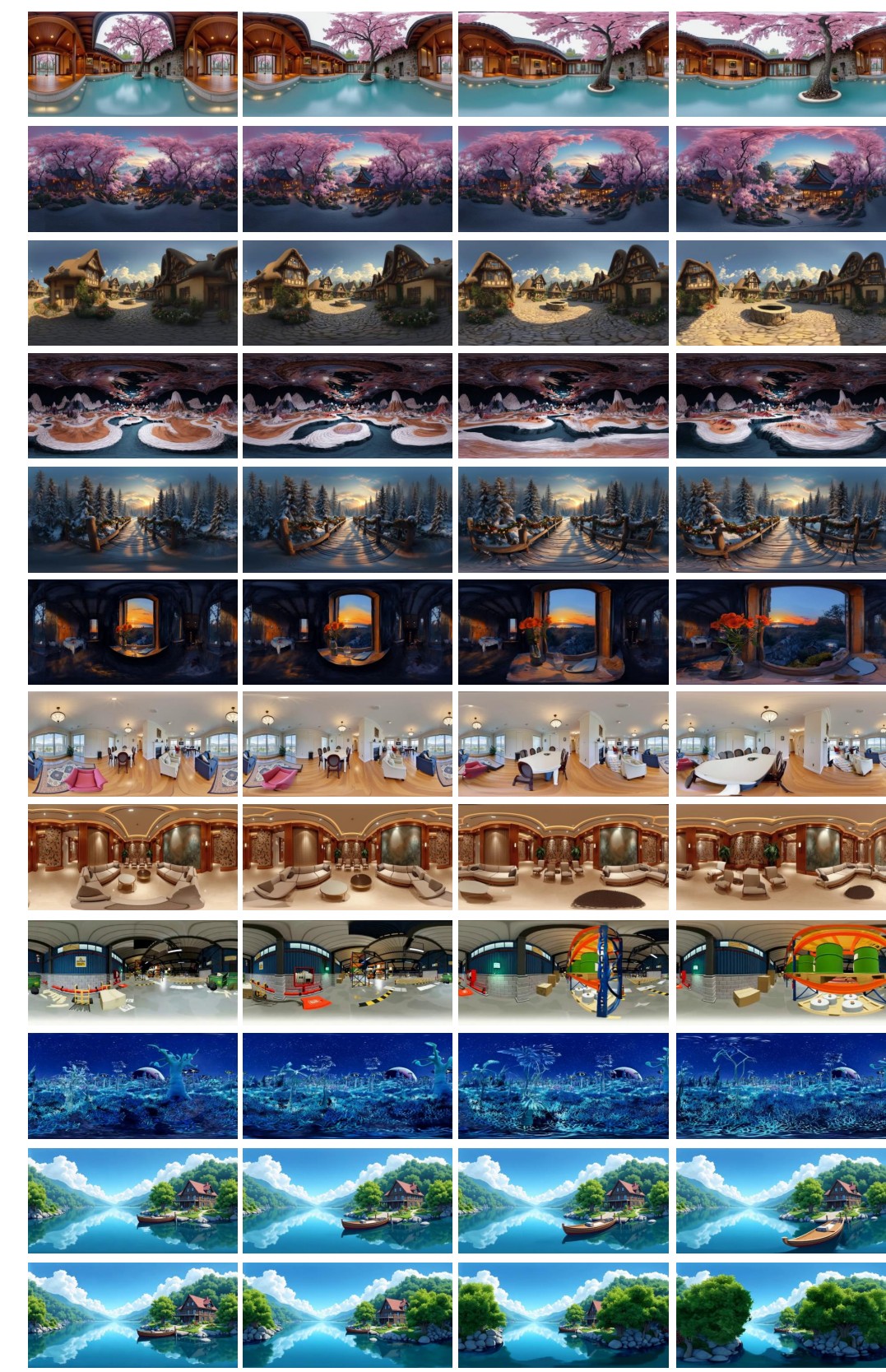

Figure B: Visualization of zero-shot generalization on in-the-wild images.

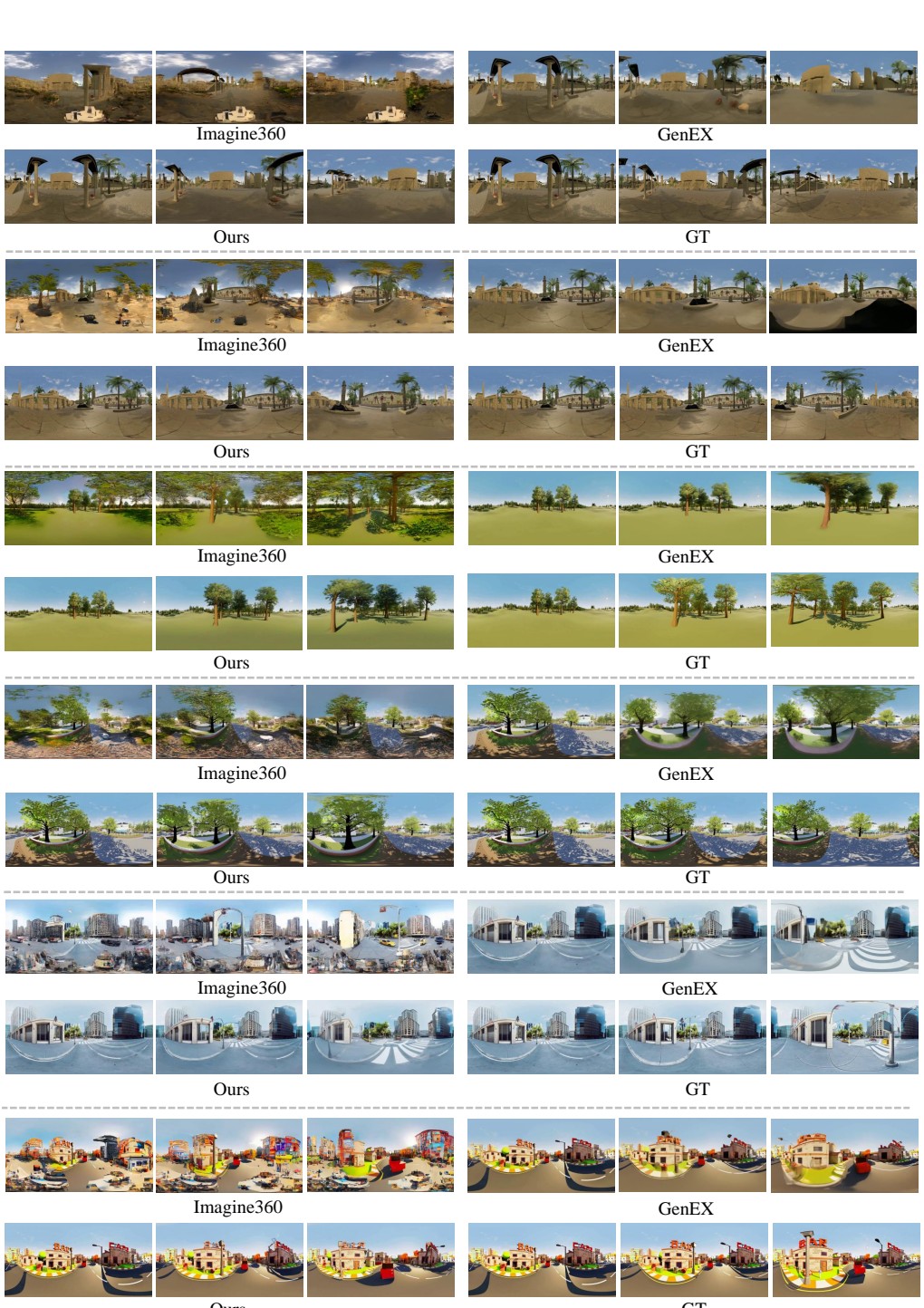

Figure C: Visualization of the comparison with panoramic generation methods on the test sets.

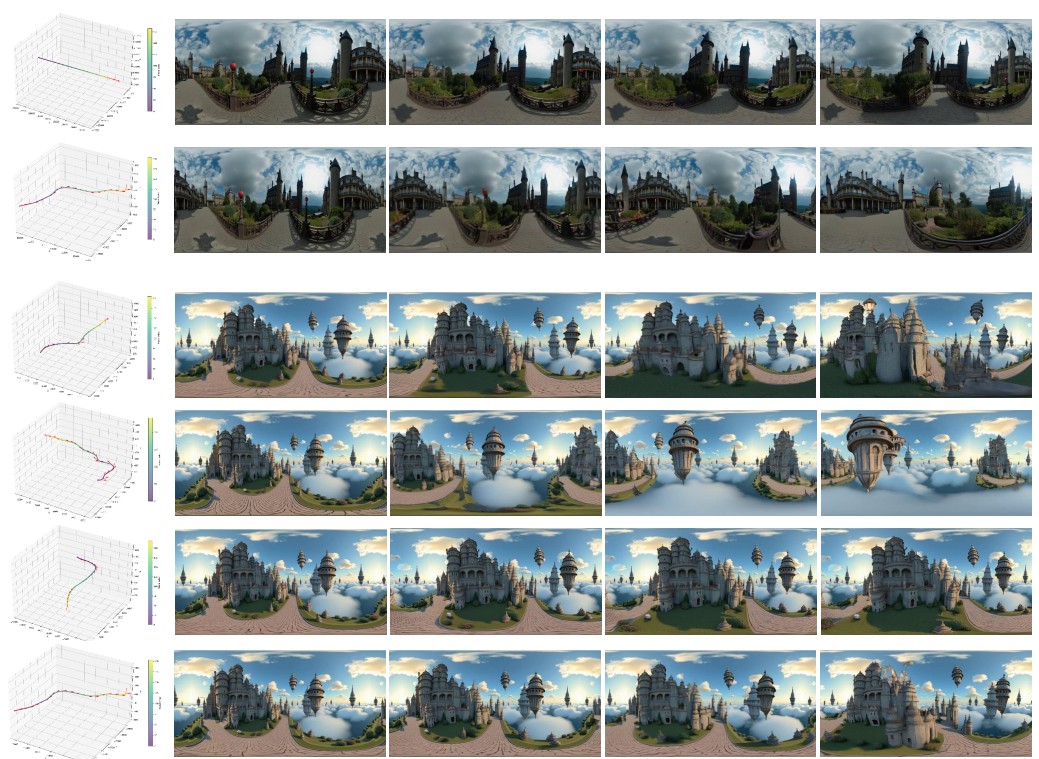

Figure D: Visualization of various exploration routes on a single image.

### D.3 VISUALIZATION OF DIVERSITY EXPLORATION ROUTES

In this section, as shown in Fig. D, we use various exploration routes on a single image to demonstrate our model's capability to handle diversity trajectory and its precise camera control functionality. All visualization results are available in **Video1-In-the-wild-results.mp4**.

## E COMPARISON WITH CAMERA-CONTROLLABLE GENERATION MODELS

In this section, we compare our method with state-of-the-art camera-controllable generation methods on test sets to assess the quality of trajectory control. Since previous methods were not trained on panoramic datasets, we crop perspective videos from panoramic videos for fair comparisons. **Due to the panoramic video resolution being 480*960, the cropped perspective videos are less than 50 pixels, yet our results maintain excellent geometric structure.**

As shown in Fig. E, our results align more closely with the ground truth compared to previous methods and demonstrate a greater range of camera motion. All visualization results are available in **Video3-Controllable-Comparison.mp4**.

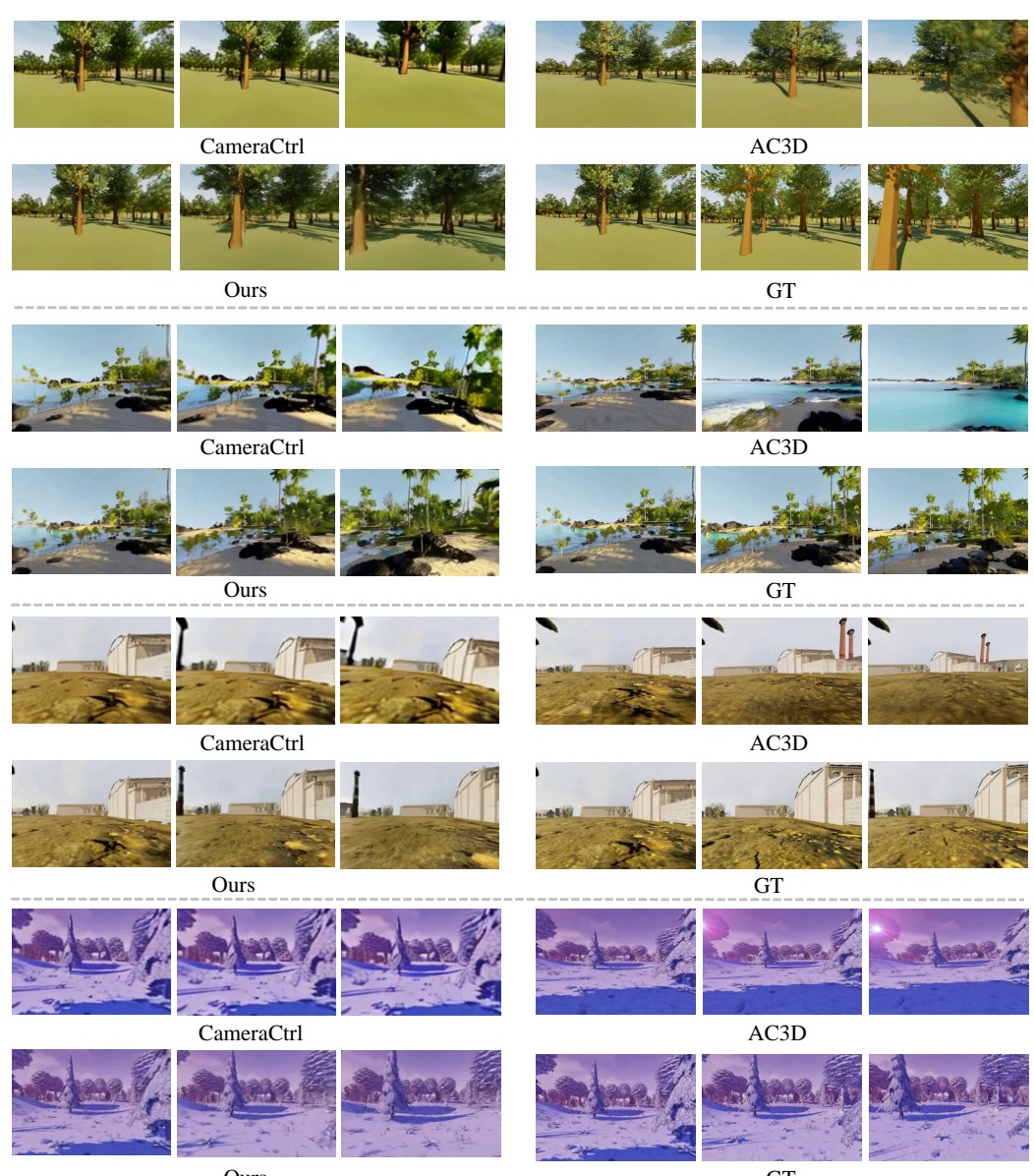

Figure E: Visualization of the comparison with camera-controllable generation methods on the test sets.

## F    FAILURE CASES AND LIMITATIONS DISCUSSION

Since our task focuses on static scene generation, we selected only static scenes during dataset construction, which typically do not include human figures. Consequently, during testing, if human figures appear in the scene, the model struggles to generate accurate panoramic warping for human figures due to the absence of such cases in the training data. Additionally, constrained by the pretrained video diffusion architecture, the current model does not support long video generation, which remains a key challenge for future research.

