# OpenReview forum: "PanoWorld-X: Generating Explorable Panoramic Worlds via Sphere-Aware Video Diffusion"
_ICLR.cc/2026/Conference — Submitted to ICLR 2026_

### Official Review · Reviewer_kKFp · 2025-10-29

**Soundness:** 3
**Presentation:** 3
**Contribution:** 3
**Rating:** 6
**Confidence:** 4

**Summary:**

This paper presents PanoWorld-X, a framework for generating controllable 360-degree panoramic videos with trajectory-guided exploration. The main contributions are: (1) PanoExplorer dataset with 116,759 panoramic video-trajectory pairs generated via Unreal Engine, (2) Explorable Sphere-Aware DiT architecture featuring Exploration-Aware Attention for trajectory control and Sphere-Aware Attention that leverages spherical geometry to improve consistency, and (3) experiments showing improvements over existing methods in visual quality, motion range, and control precision. The work enables immersive world generation for VR, embodied AI, and autonomous driving applications.

**Strengths:**

1. The PanoExplorer dataset addresses a critical gap in panoramic video research with 116,759 high-quality samples featuring diverse scenes (10 categories), extensive motion patterns, and precise trajectory annotations. The data curation pipeline is well-designed with collision detection and multi-stage quality filtering.
2. The sphere-aware attention mechanism is theoretically well-motivated, properly accounting for the spherical topology of panoramic data.
3.  The paper provides extensive comparisons with both panoramic generation methods (360DVD, Imagine360, GenEX) and camera-controllable methods (CameraCtrl, AC3D), demonstrating clear improvements across multiple metrics. Ablation studies systematically validate each component.

**Weaknesses:**

1. The entire dataset is synthetic from Unreal Engine. While Section D.1 shows some zero-shot generalization to in-the-wild images, more rigorous evaluation on real panoramic videos would strengthen the claims about practical applicability. The domain gap could limit real-world deployment.
2. There lacks some analyze regarding the proposed datasets. For example, this paper does not give the camera trajectory distribution (move forward, move forward and turn left / right, etc), which is important to understand the proposed datasets. And keeping a balanced camera trajectory is important for learning a model with good trajectory generalization ability.
3. The comparison between PanoWorld-X and other methods is unfair, since other methods are not trained using the proposed datasets, while the evaluation dataset comes from the proposed dataset.

**Questions:**

1. How to define the yaw, pitch, and roll angles (Line 252)?
2. Can the author give more detailed description regarding the scale normalization strategy? (Line 263)
3. Can the author compare PanoWorld-X with other baseline methods using the datasets comes from other sources?

---

> ### Author Response · Authors · 2025-11-21
>
> Dear Reviewer kKFp, thank you very much for taking the time to review our work and for providing constructive feedback. We sincerely appreciate your acknowledgment of our paper, which provides a valuable dataset for future research, well-motivated modules, and strong performance surpassing previous models. We would like to take this opportunity to address your comments.
>
> ***
>
> **Q1.Real Panoramic Video Performance**
>
> To evaluate our model’s performance on real-world data, we select 100 cases from the WEB360 dataset, along with additional real-world panoramic images collected by ourselves to maximize the diversity of scene categories. The results show that our model achieves lower FID and FVD scores, indicating that fine-tuning on synthetic data can effectively adapt large-scale pre-trained weights to the panoramic modality and yield strong performance on real-world data. Moreover, the higher Q-Align scores further demonstrate that our method is also better under large-model-based evaluation.
>
> | Model      | FID   |  FVD    | Q-Align |
> |----------------|---------|----------------|----------------|
> | 360DVD    | 92.31  |     167.39   |  0.487   |
> | Imagine360 | 87.24  |     135.03   |  0.492   |
> | GenEX   | 82.31   |    142.35     |  0.413   |
> | Ours    |  **64.82**     | **103.45**   |    **0.551**     |
>
>
> ***
>
> **Q2.Datasets Analysis**
>
> Thanks for your attention to our proposed datasets. We have provided detailed statistics of our datasets in Supplementary Section B,(which can be found after the references in the main paper, or in appendix.pdf in the submitted zip). This section includes visualization examples, scene category statistics, trajectory complexity statistics, the caption annotation pipeline with examples, as well as a comparison with existing panoramic datasets.
>
> Your comment on trajectory balance is very insightful. We were aware of this issue during dataset construction and deliberately designed for diversity. Concerning trajectory diversity, we report detailed statistics in the supplementary material. We use tortuosity (defined as the ratio between the straight-line distance and the actual path length between the start and end points) to quantify trajectory complexity. As shown in Table B in the supplementary material, only a small portion of trajectories are nearly perfectly straight; the majority contain at least one or two noticeable directional changes, and the remaining trajectories exhibit rapid turns. This distribution indicates that our trajectories are reasonably diverse rather than being dominated by simple linear paths.
>
> | Tortuosity Range     |  Percentage   | Meaning Interpretation  |
> |----------------|---------|----------------|
> | 1                   | 7.13%   | Perfectly straight path    |
> | 1>T≥0.75      | 60.24%   | Noticeable directional changes, possibly with one or two turns      |
> | 0.75>T ≥0.5     | 32.29%   | Significant directional changes, possibly with multiple turns       |
> | 0.5>T ≥0     | 0.34%   | Highly complex trajectories          |
>
> To further analyze the types of trajectories, as suggested by the reviewer, we categorize them into six canonical patterns: (1) Straight-Line, (2) Turn Left, (3) Turn Right, (4) Alternating-Turn (Z-shaped, S-shaped), (5) Closed-Loop (curve-shaped), and (6) Round-Trip (U-shaped). We determine the trajectory category by analyzing the curvature and its direction along the motion path. The statistics for these categories are reported in the following table, and they demonstrate that our trajectories exhibit both substantial complexity and diversity.
>
>
> | types of trajectories    |  Percentage   |
> |----------------|---------|
> | Straight-Line               | 12.45%   |
> | Turn Left     |28.71%   |
> | Turn Right     | 24.53%   |
> | Alternating-Turn (Z-shaped, S-shaped)  | 14.34%   |
> | Closed-Loop (curve-shaped)   | 10.91%   |
> | Round-Trip (U-shaped)   | 9.06%   |
>
>
> ***

---

> > ### Author Response · Authors · 2025-11-21
> >
> > **Q3.Training Other Models on the Proposed Datasets**
> >
> > To enable a fair comparison between previous methods and ours, we train other models on the proposed datasets. Since Imagine360 has not released its training code, we fine-tune 360DVD and GenEX on our datasets instead. The results are as follows: both models achieve improved performance after fine-tuning, with particularly notable gains in FID and FVD. However, their overall performance still does not surpass our models.
> >
> > | Model      | PSNR    |  SSIM |  LPIPS   | FID |  FVD  |
> > |----------------|---------|----------------|----------------|----------------|----------------|
> > | 360DVD    | 10.66   |     0.35    |  0.73   |   111.85     | 2049.21  ｜
> > | 360DVD(finetuned)  | 13.31   |     0.48   |  0.68   |   79.32    | 1239.13 ｜
> > | GenEX   | 16.12   |     0.59    |  0.42   |   42.22    | 1113.72  ｜
> > | GenEX(finetuned)  | 17.29  |     0.61   |  0.36   |   36.99   | 668.23｜
> > | Ours    | **19.34**   |     **0.63**     | **0.24**   |    **28.01**     |**467.18**  ｜
> >
> >
> > ***
> >
> >
> >
> > **Q4.Definition of Yaw, Pitch, and Roll Angles**
> >
> > The input signals driving the camera movement are derived from Unreal Engine, which provides a six-degree-of-freedom (6-DoF) control signal $Er_i$ for camera $i$:
> > $$Er_i = (x_i, y_i, z_i, \alpha_i, \beta_i, \gamma_i)$$
> > This signal represents the camera's spatial coordinates $(x_i, y_i, z_i)$ and its orientation, which is uniquely determined by the Euler angles $(\alpha_i, \beta_i, \gamma_i)$ representing rotations around the Z, Y, and X axes, named as yaw,pitch and roll respectively. For rotation $R$ of extrinsic matrix, it is constructed from the basic single-axis rotation matrices. For example, the rotation around the Z-axis by angle $\alpha$ is:
> > $$
> > R_z(\alpha) =
> > \begin{bmatrix} \cos\alpha & -\sin\alpha & 0 \end{bmatrix} \\
> > \begin{bmatrix} \sin\alpha & \cos\alpha & 0 \end{bmatrix} \\
> > \begin{bmatrix} 0 & 0 & 1 \end{bmatrix}
> > $$
> > ($R_z$ is a 3×3 matrix, which cannot be displayed correctly in OpenReview)
> > By sequentially multiplying the basic rotation matrices in the Z-Y-X order, the final rotation matrix $R$ is determined as $R = R_x(\gamma) R_y(\beta) R_z(\alpha)$.
> >
> > ***
> >
> > **Q5.Details on Scale Normalization**
> >
> > We introduce a new normalization strategy to address the highly varying absolute scales across different environments. For example, outdoor scenes such as forests or roads typically span much larger physical spaces than indoor scenes. As a result, the same amount of camera movement produces only minor visual changes outdoors, whereas it causes much more noticeable changes indoors. Without proper normalization, representing outdoor motion would require substantially larger values of $(\Delta x, \Delta y, \Delta z)$ leading to a highly complex and uneven data distribution and ultimately hindering training efficiency. To mitigate this issue, we select the 75th percentile of metric depth (obtained from Unreal Engine) as an anchor scale, set to 10 cm. This approach effectively rescales all scenes to a common reference scale. We then normalize all displacement values to the range [0,1] within each clip, so that the motion magnitude is represented consistently regardless of the scene. In other words, the normalization strategy adaptively scales motion according to the scene’s absolute size. For example, a normalized displacement of 0.5 corresponds to a larger absolute movement in an outdoor scene, than in an indoor scene, ensuring more substantial visual changes.
> > This strategy is validated by our ablation study (Line 449). By removing the impact of scene-dependent absolute scale, the complexity of motion trajectories in the dataset is reduced, resulting in higher visual evaluation metrics and improved control accuracy.

---

> > > ### Comment · Reviewer_kKFp · 2025-11-26
> > >
> > > The author rebuttal has addressed most of the concerns. There is another one question regarding the evaluation set.
> > > When ablating the effectiveness of scale normalization, were the ground truths of rotation error and translation error also normalized?

---

> > > > ### Author Response · Authors · 2025-11-27
> > > >
> > > > Thank you for taking the time to review our work and for recognizing our efforts.
> > > >
> > > > Regarding scale normalization in the evaluation: since rotation is not affected by scale, we do not normalize the rotation parameters and instead use the original rotations as the ground truth. For translation,  the normalized values are treated as the new ground truth. Conceptually, this means that we do not use the real-world scale as the reference; instead, we evaluate everything in the rescaled space, so that scenes with different original scales are brought to a comparable level.
> > > >
> > > > We hope our response addresses your concerns.

---

> > > > > ### Comment · Reviewer_kKFp · 2025-11-27
> > > > >
> > > > > That make scenes. I have no more questions, and will raise my rating.

---

> > > > > > ### Author Response · Authors · 2025-11-27
> > > > > >
> > > > > > We are sincerely grateful that you raised the rating. Your positive evaluation and recognition of this paper’s contributions are very encouraging for us.

---

### Official Review · Reviewer_3MJN · 2025-10-31

**Soundness:** 3
**Presentation:** 3
**Contribution:** 3
**Rating:** 4
**Confidence:** 4

**Summary:**

The paper proposes PanoWorldX, a novel framework for high-fidelity and controllable panoramic video generation with diverse camera trajectories. PanoWorldX consists of a novel pipeline using the Unreal Engine to procedurally generate a large, diverse dataset of panoramic video-trajectory pairs, and a Sphere-Aware Diffusion Transformer that reprojects equirectangular features onto the spherical surface to accurately model geometric adjacency in the latent space. By specifically addressing the geometric misalignment between the spherical geometry of panoramic data and conventional diffusion model priors, PanoWorldX demonstrates significant improvements over existing methods in terms of motion range, control precision, and visual quality.

**Strengths:**

- The panorama data synthesis pipeline for large-scale and accurately labeled panoramic video-trajectory pairs is a valuable contribution, as the scarcity of data has limited the development of this task.
- The Sphere-Aware Attention and Exploration-Aware Attention are theoretically sound and practical for integration into existing DiT architectures. Explicitly modeling for the spherical topology is effective for high-fidelity panoramic generation,
- Experimental results demonstrate that PanoWorldX outperforms baseline methods and achieves good panoramic video generation performance.

**Weaknesses:**

- While I agree that synthesized panoramic data is necessary, the domain gap between synthetic and real-world data remains significant. Complex real-world scenarios, such as variations in lighting, non-rigid deformation, and motion blur and noise discrepancies that occur during shooting, remain challenging to represent in synthetic data. This paper's analysis of the data lacks a detailed discussion of these cases, which may be beneficial for future high-quality panoramic data synthesis.
- PanoWorldX focuses heavily on static scene generation and camera movement. It is unclear how PanoWorld-X handles highly dynamic, non-structural elements, such as moving characters, weather changes, or complex large motions. Further discussion or ablation studies on the temporal stability of non-rigid objects would enhance the review of temporal coherence.

**Questions:**

- Modifying the standard DiT architecture with additional attention branches (Sphere-Aware and Exploration-Aware) inherently increases computational load during training and inference. What is the latency or memory overhead compared to the baseline DiT model?
- While trajectory complexity was discussed, can the model handle cases like abrupt changes in elevation, sharp turns in confined spaces, or rapid changes in camera velocity without noticeable artifacts?
- What is the individual contribution of the Exploration-Aware Attention in the ablation study?

---

> ### Author Response · Authors · 2025-11-21
>
> Dear Reviewer 3MJN, We sincerely appreciate the time and effort you have dedicated to reviewing our work and providing valuable feedback. We are also grateful for your recognition of the merits of our datasets, the proposed sphere-aware module, and the promising experimental results. After carefully considering your comments, we are pleased to address each of your concerns in detail as follows:
>
>
> **Q1.Differences between synthetic data and real-world data**
>
> We appreciate the reviewer’s concern about the domain gap between synthetic and real-world panoramic data.We fully agree that complex real-world phenomena are difficult to synthesize perfectly.
>
> **(1)In our work, we have tried to leverage Unreal Engine’s simulation capabilities to approximate real data as much as possible.** In addition to the basic rendering pipeline, our synthetic panoramas are enhanced with Unreal Engine’s time-of-day and weather systems. For a subset of scenes, we explicitly vary the simulated time (e.g., morning vs. evening) and illumination conditions (e.g., sunny vs. overcast). These settings jointly affect global illumination, shadow directions and lengths, and the color temperature of light across the entire panorama. Combined with UE’s physically based rendering and sky/atmospheric models, this allows us to approximate a range of complex lighting effects such as soft shadows, indirect bounces, and spatially varying color shifts. We also exploit UE’s post-processing pipeline to introduce camera effects such as exposure changes and motion blur, so that the synthesized panoramas are not purely “clean” renderings but incorporate certain aspects of the real image formation process.
>
> **(2)Nevertheless, we fully acknowledge that, with the current capabilities of game-engine, certain real-world phenomena are still difficult to reproduce faithfully in synthetic panoramas.**  For example, even with UE it remains very challenging to accurately capture sensor- and device-specific noise patterns, rolling-shutter artifacts, strong hand-held camera shake, or compression artifacts, and the diversity of non-rigid deformations (e.g., complex human motions, clothing and hair dynamics) is still more limited than in unconstrained real scenes. These factors are intrinsically difficult to synthesize in a physically accurate and computationally feasible manner with today’s tools, and thus reflect broader technological limitations rather than a design choice specific to our framework. We expect these gaps to be gradually alleviated as rendering engines, sensor models, and generative techniques continue to advance, and extending our framework to better capture these effects is an important direction for future work.
>
> **(3)Finally, our main contribution is to provide the first large-scale panoramic video synthesis framework with diverse scene types and accurately paired camera trajectories.** Compared with prior synthetic datasets such as DiffPano (NeurIPS 2024), which are restricted to indoor scenarios, and real-captured datasets such as Web360 (CVPR 2024) and Imagine360 (NeurIPS 2025), our dataset covers a broader range of environments and larger motion amplitudes, while offering absolutely accurate camera trajectory annotations that real data cannot easily provide. We believe that both our data-generation pipeline and the resulting dataset will be valuable resources for future research.

---

> > ### Author Response · Authors · 2025-11-21
> >
> > **Q2.Dynamic Case Performance**
> > We thank the reviewer for this insightful question regarding dynamic scenes. We respond in two aspects:
> >
> > **(1) Technical bottlenecks of dynamic panoramic data.** The temporal stability of non-rigid objects depends not only on network design but also on the construction of appropriate datasets. Regarding moving objects, we deliberately focus on static scenes because the technology for generating high-fidelity dynamic synthetic panoramas is still immature. Key technical challenges remain unsolved, such as inserting objects into large-scale virtual environments [1,2], rigging them for animation [3,4], driving physically plausible motion [5,6,7], and ensuring that these motions follow high-level semantic requirements [8,9]. Relying on these immature synthetic pipelines would likely introduce significant noise and artifacts into the training data. Another way is real-world capture, which is extremely labor-intensive, difficult to scale, and often infeasible in complex environments such as glaciers or cliffs where filming is unsafe or impossible.  **As a result, our current model is inherently constrained by the lack of a large-scale dynamic panoramic dataset, and we do not claim strong performance on highly dynamic content.** We explicitly discuss this limitation and the corresponding failure cases in the FAILURE CASES AND LIMITATIONS DISCUSSION section (Sec.F in supplementary material ). We therefore treat robust handling of moving objects as a natural next step for future work, as synthetic data generation tools and dynamic panoramic resources continue to evolve.
> >
> > **(2) Main contribution of the static synthetic dataset pipeline and controllable generation.** We emphasize that our constructed dataset is, to the best of our knowledge, the first large-scale and diverse synthetic panoramic dataset specifically designed for trajectory control. To address the core challenges of controllable panoramic video generation and preserving spherical projection consistency, it is crucial to first disentangle camera trajectory motion from object motion. Existing synthetic and real-world panoramic datasets either lack accurate trajectory annotations or provide limited scene diversity. Our dataset synthesis pipeline and the curated data therefore fill this gap and offer a solid foundation for future research on trajectory-controllable panoramic video generation and other panoramic-related tasks.
> >
> >
> >
> >
> >
> >
> > [1]NeRF-Insert: 3D Local Editing with Multimodal Control Signals
> >
> > [2]TIP-Editor: An Accurate 3D Editor Following Both Text-Prompts And Image-Prompts
> >
> > [3]MoDA: Modeling Deformable 3D Objects from Casual Videos
> >
> > [4]MagicArticulate: Make Your 3D Models Articulation-Ready
> >
> > [5]Stable Part Diffusion 4D: Multi-View RGB and Kinematic Parts Video Generation
> >
> > [6]Puppeteer: Rig and Animate Your 3D Models
> >
> > [7]Relightable and Animatable Neural Avatar from Sparse-View Video
> >
> > [8]FantasyHSI: Video-Generation-Centric 4D Human Synthesis In Any Scene through A Graph-based Multi-Agent Framework
> >
> > [9]InteractMove: Text-Controlled Human-Object Interaction Generation in 3D Scenes with Movable Objects

---

> > > ### Author Response · Authors · 2025-11-21
> > >
> > > **Q3. Computational Cost Analysis**
> > >
> > > Indeed, our additional branch increases both memory usage and computation, which is essentially a trade-off between performance and cost. We conducted a quantitative evaluation: the original CogVideoX-5B-I2V requires 5G of memory for inference and about 200s on a single A800 GPU to generate a 49-frame video, while our model requires 8G of memory and about 280s.The original model LoRA fine-tuning model is 63 GB, while our model is 78 GB. We believe this trade-off is worthwhile, because it not only improves visual detail and quantitative metrics, but also enhances the consistency of spherical projection on the panoramic sphere, especially the left-right consistency (as shown in Supplementary Video4-ablation-sphere-attn.mp4). This is crucial for panoramic applications, since on head-mounted displays or when converting to a perspective view, visible vertical seams at the left and right boundaries can significantly degrade the immersive experience.
> > >
> > > ***
> > >
> > > **Q4.Hard Cases with Complex Trajectories**
> > >
> > > Thanks for your advice. We have shown some complex cases in video 1-in-the-wild-results.mp4.
> > >
> > > As for sharp turns, in the playground case (17s-20s), we initially shifted from the path directly ahead to the path on the left. In the Equirectangular Projection (ERP), this appears as a pixel movement across half the width. However, in the corresponding perspective view (normal human vision), due to the nature of the projection, this actually represents a 180-degree rotation, effectively turning from directly forward to directly backward. In the garden example (3s-6s), the same principle applies: the panoramic image first turns left across nearly one-quarter of the width, then turns right by one-quarter of the width. In the perspective view, this is equivalent to rotating 90 degrees left and then 90 degrees right. Our other examples feature similar camera trajectory settings. We believe that rotations of around 90-180 degrees in the physical world sufficiently demonstrate the capability to handle sharp turns. Regarding larger turns, such as a massive rotation in ERP projection corresponding to over 180 degrees (approaching 270 degrees) in the real world, these do not align with the requirements and distribution of common real-world scenarios.
> > >
> > > As for abrupt changes in elevation, we intentionally did not include cases of abrupt elevation changes during dataset preparation for two main reasons: (1) Due to the nature of Equirectangular Projection, a single image covers a 360-degree field of view. Therefore, even with an eye-level view, one can directly convert 2D ERP pixels to 3D spherical coordinates, apply a rotation matrix around the axis, and re-project back to the 2D plane. This transformation process can be achieved via post-processing. Consequently, any generated eye-level view can be transformed into a view with freely specified elevation through simple mathematical transformations. (2) Although our dataset is the first large-scale, diverse, trajectory-paired dataset (exceeding 116K samples), it is still relatively small compared to natural image datasets. Introducing degrees of freedom for elevation would significantly increase data complexity, making training more difficult and convergence harder to achieve. Given that this can be addressed mathematically as mentioned in point (1), we believe it should not be implemented through a more complex generation approach.
> > >
> > > As for camera velocity changes, we did not intentionally create a dataset with varying movement trajectory speeds during preparation. However, we adopted an adjusted sampling strategy during training. For example, each generated video consists of 49 frames; for a source video of 300 frames, we can set a random sampling interval (stride) of 1-6 frames. With sufficient training epochs, videos sampled at different random intervals can be regarded as content captured at different camera speeds. Based on this sampling setting rather than a dataset-specific setting, we acknowledge that the model possesses some capability for velocity changes, though it is not extremely sensitive. For instance, the example at 48s-51s shows fast-then-slow movement, while the case at 45s-48s shows slow-then-fast movement. Our cases reflect the ability to represent camera velocity changes. Of course, preparing a dataset specifically for these conditions would enhance the intensity of these variations.

---

> > > > ### Author Response · Authors · 2025-11-21
> > > >
> > > > **Q5.Ablation study about Exploration-Aware Attention**
> > > >
> > > > Our exploration-aware attention includes an exploration route representation and a controllable branch. The main contribution lies in designing control signals tailored to the specific properties of panoramic data. Since research on control injection modules is already well-established, we adopted a design based on ControlNet with subtle modifications. The main difference is that we feed both the video latent and the condition latent into the controllable branch, rather than using only the condition latent. Based on this information, we have supplemented our verification with ablation studies.
> > > >
> > > > **Ablation without exploration route representation.** The original movement commands consist of 6 Degrees of Freedom (6-DoF), denoted as $E_r = (x, y, z, \alpha, \beta, \gamma)$, which can be directly sent into the module via an encoder. However, 6-DoF cannot provide detailed trajectory control information to guide the directional movement of the visual content; therefore, we aimed to design a per-pixel level guidance signal. Previous methods have demonstrated that Plucker Embedding is a more reasonable approach; however, due to the lack of intrinsic and extrinsic information in panoramic images, we designed an equivalent transformation based on panoramic data characteristics (Lines 244-262). The second design utilizes a depth/scale normalization strategy (Lines 262-268) to reduce the highly varying absolute scales of the original signals. The ablation study demonstrates that the proposed exploration improves image quality metrics and, most importantly, substantially enhances control precision, which is primarily reflected in $R_{err}$ and $T_{err}$. The main reason for this is that the information contained in the 6-DoF is too sparse, and directly encoding it into the latent space lacks precise control over the trajectory.
> > > >
> > > > | Model      | PSNR    |  SSIM |  LPIPS   | FID |  FVD |$ R_{err}$ | $T_{err}$ |
> > > > |----------------|---------|----------------|----------------|----------------|----------------|----------------|----------------|
> > > > | w/o exploration route representation | 18.01   |     0.52     | 0.40   |     39.58     |544.95   |     0.109  |  0.192   |
> > > > | Full Model    | 19.34   |     0.63     | 0.24  |    28.01     | 467.18   |  0.061     |  0.073   |
> > > >
> > > >
> > > >
> > > > **Ablation without the concatenation of the video latent and the condition latent.**
> > > > While the original ControlNet only feeds the condition latent into the controllable branch, we concatenate the video latent and the condition latent to feed into the branch together. The underlying rationale is that, unlike conditions such as sketches or poses, camera information in our task is not explicit; therefore, providing the network with video content helps it better interpret the control signals. The ablation study demonstrates that this strategy yields improvements across various metrics.
> > > >
> > > > | Model      | PSNR    |  SSIM |  LPIPS   | FID |  FVD |$ R_{err}$ | $T_{err}$ |
> > > > |----------------|---------|----------------|----------------|----------------|----------------|----------------|----------------|
> > > > | w/o concatentation | 17.87   |     0.52     | 0.34   |     40.21     |501.03   |     0.082  |  0.134   |
> > > > | Full Model    | **19.34**   |     **0.63**     | **0.24**   |    **28.01**     |**467.18**   |  **0.061**     |  **0.073**   |

---

### Official Review · Reviewer_8Add · 2025-11-01

**Soundness:** 3
**Presentation:** 2
**Contribution:** 3
**Rating:** 4
**Confidence:** 4

**Summary:**

1. The paper introduces PanoWorld-X, a framework for generating controllable 360-degree panoramic videos with diverse exploration paths. 2. The paper proposes the PanoExplorer dataset, a large-scale synthetic dataset of panoramic video-trajectory pairs.
3. The paper proposes sphere-aware DiT block. This block consists of explorable-aware attention to handle trajectory control and a sphere-aware attention to improve spatiotemporal consistency.
4. The experimental results demonstrate that PanoWorld-X achieves superior performance compared to previous panoramic video generation and trajectory-based methods.

**Strengths:**

1. The paper introduces PanoWorld-X, which can generate high-fidelity, explorable panoramic videos controlled by trajectories.
2. The proposed Sphere-Aware DiT is the main technical contribution. It is well-motivated by the need to address the geometric misalignment between spherical panorama data and standard diffusion models, explicitly capturing spherical connectivity.
3. The PanoExplorer dataset is a valuable contribution to the field, which contains 110k panoramic videos paired with trajectory annotations from Unreal Engine.

**Weaknesses:**

1. The architectural contribution of the trajectory control mechanism(ControlNet) appears limited and there are many previous works about it in the field of general video generation. Another novelty is sphere-aware attention, where the similar spherical representation usage is discussed in previous panoramic image generation works likes PanFusion and Text2Light.
2. The explanation of the Exploration Route Representation in Section 3.3 lacks clarity. The paper introduces a 6-DoF signal $Er_i$​ but then discusses converting it into extrinsic matrices and using Plücker embeddings. The precise relationship between this initial 6-DoF vector and the final $R^{F×H×W×6}$ embedding tensor depicted in Figure 2 is ambiguous and needs to be clarified.
3. The practical utility of the framework is severely constrained by its low output resolution of 480×960. For panoramic content, this resolution is insufficient for reality usage, especially given the base model supports higher resolutions. This issue is acknowledged in the supplementary materials, where perspective crops are described as being "less than 50 pixels".
4. While quantitative metrics on the evaluation of trajectory adherence are provided, the paper do not offer side-by-side visualization of the desired camera path overlaid on the generated video to demonstrate the model's trajectory-conditioned effect on the generated video.

I will consider adjusting my score if my concerns are well-resolved.

**Questions:**

1. Given the final generated resolution is 480×960, what is the native resolution of the PanoExplorer dataset? It connects with the utility of the dataset for future high-resolution panoramic research.
2. How capable is the model of inferring an exploration path purely based on text instructions?
3. The current evaluation focuses heavily on static, synthetic scenes. How does PanoWorld-X handle out-of-distribution real-world panoramic images even with moving objects?

---

> ### Author Response · Authors · 2025-11-21
>
> Dear Reviewer 8Add, thank you for taking the time to review our work and for providing constructive feedback. We sincerely appreciate your recognition of our pipeline and the roles of its components, as well as your interest in our datasets and acknowledgement of their value to the community. We have carefully considered your suggestions and are committed to addressing all of your concerns as outlined below.
> ***
>
> **Q1.Novelty about Trajectory Control Mechanisms**
>
> We acknowledge that many excellent works have recently emerged in controllable video diffusion. These studies have already provided comprehensive explorations of control mechanisms that integrate various forms of control signals with the main diffusion network. **For subsequent domain-specific work, designing control signals according to the task is often more important—and more challenging—than designing the network modules that inject them. Therefore, our innovation in *Exploration-Aware Attention* focuses primarily on the *representation of the exploration route*, rather than on how control signals are injected into the main branch.** We apologize for not making this distinction clear in our manuscript, which may have led to your misunderstanding.
>
> Specifically, there is no prior work on how to control panoramic video, making it challenging to design control signals that respect the unique properties of panoramic data. As reported in Lines 244–268, we summarize two main challenges.
>
> **First, we address how to convert 6-DoF movement commands into the equivalent intrinsic and extrinsic parameters used in perspective projection.** Previous view-controllable video generation methods typically rely on intrinsic and extrinsic matrices to compute Plücker Embeddings for view guidance. However, panoramic data does not naturally possess the notion of intrinsic or extrinsic parameters, which makes Plücker Embedding computation infeasible. To overcome this, we convert the relative spatial coordinates$(\Delta x, \Delta y, \Delta z)$(with respect to the first frame) into a translation matrix $T$, and transform the viewing angles$(\Delta \alpha, \Delta \beta, \Delta \gamma)$ into a rotation matrix $R$. Together, these form the extrinsic parameters. The detailed explanation is provided in our response to Q2. Regarding the intrinsic parameters, panoramic cameras use a spherical projection model rather than a planar imaging model, therefore they do not have a standard intrinsic matrix. To enable Plücker Embedding, we adopt a minimal intrinsic assumption, which we also explain in detail in Q2.
>
> **Second, we introduce a new normalization strategy to address the highly varying absolute scales across different environments.** For example, outdoor scenes such as forests or roads typically span much larger physical spaces than indoor scenes. As a result, the same amount of camera movement produces only minor visual changes outdoors, whereas it causes much more noticeable changes indoors. Without proper normalization, representing outdoor motion would require substantially larger values of $(\Delta x, \Delta y, \Delta z)$ leading to a highly complex and uneven data distribution and ultimately hindering training efficiency. To mitigate this issue, we select the 75th percentile of all depth as an anchor scale (set to 10 cm) and normalize all displacement values to the range [0,1] within each clip. This strategy is validated by our ablation study (Line 449), which demonstrates its effectiveness.
>
>
> Based on the above analysis and design, we obtain motion signals that represent the panoramic trajectory. For injecting control signals into the main branch, we follow the operations used in ControlNet. **The main difference is that we feed both the video latent and the condition latent into the controllable branch, rather than using only the condition latent.** The underlying rationale is that, unlike conditions such as sketches or poses, camera information in our task is not explicit; therefore, providing the network with video content helps it better interpret the control signals. The ablation study demonstrates that this strategy yields improvements across various metrics.
>
>
> | Model      | PSNR    |  SSIM |  LPIPS   | FID |  FVD |$R_{err}$ | $T_{err}$ |
> |----------------|---------|----------------|----------------|----------------|----------------|----------------|----------------|
> | w/o concatentation | 17.87   |     0.52     | 0.34   |     40.21     |501.03   |     0.082  |  0.134   |
> | Full Model    | **19.34**   |     **0.63**     | **0.24**   |    **28.01**     |**467.18**   |  **0.061**     |  **0.073**   |
>
>
> ***

---

> > ### Author Response · Authors · 2025-11-21
> >
> > **Q2.Details of Route Exploration Representation**
> >
> > Our goal is to design a representation that effectively expresses view movement in panoramic video. We choose Plücker Embeddings as this representation because they provide pixel-wise positional information and enable stronger control capabilities than directly using camera parameters. This advantage has been demonstrated in the ablation study of CameraCtrl[1].  Plücker Embeddings consist of intrinsic and extrinsic parameters. However, as explained in Q1, panoramic data does not naturally provide such parameters, so we must convert them into an equivalent form that matches the characteristics of panoramic data.
> >
> > - **Origin Signal:**
> > The input signals driving the camera movement are derived from Unreal Engine, which provides a six-degree-of-freedom (6-DoF) control signal $Er_i$ for camera $i$:
> > $$Er_i = (x_i, y_i, z_i, \alpha_i, \beta_i, \gamma_i)$$
> > This signal represents the camera's spatial coordinates $(x_i, y_i, z_i)$ and its orientation, which is uniquely determined by the Euler angles $(\alpha_i, \beta_i, \gamma_i)$ representing rotations around the Z, Y, and X axes, respectively.
> >
> > - **Extrinsic:**
> > The camera's extrinsic parameters are defined by the Translation Matrix $T$ and the Rotation Matrix $R$. For translation $T$, we use the relative spatial coordinates $(\Delta x, \Delta y, \Delta z)$ calculated with respect to the first frame's position to represent the displacement from the world coordinate center. For rotation $R$, it is constructed from the basic single-axis rotation matrices. For example, the rotation around the Z-axis by angle $\alpha$ is:
> > $$
> > R_z(\alpha) =
> > \begin{bmatrix} \cos\alpha & -\sin\alpha & 0 \end{bmatrix} \\
> > \begin{bmatrix} \sin\alpha & \cos\alpha & 0 \end{bmatrix} \\
> > \begin{bmatrix} 0 & 0 & 1 \end{bmatrix}
> > $$
> > ($R_z$ is a 3×3 matrix, which cannot be displayed correctly in OpenReview)
> > By sequentially multiplying the basic rotation matrices in the Z-Y-X order, the final rotation matrix $R$ is determined as $R = R_x(\gamma) R_y(\beta) R_z(\alpha)$.
> >
> > - **Intrinsic:**
> > To obtain a minimal intrinsic matrix for equirectangular panoramic images, we convert the angular sampling into an equivalent focal length in pixels while incorporating standard simplifying assumptions, specifically zero distortion and setting the principal point at the image center. Given a panorama of width $W$ and height $H$, the horizontal angular resolution, $\Delta\theta$, is defined as:
> > $$\Delta\theta = \frac{2\pi}{W}$$
> > We then define the focal length in pixels as the inverse of this angular resolution, yielding identical values for both the horizontal ($f_x$) and vertical ($f_y$) focal lengths:
> > $$f_x = f_y = \frac{W}{2\pi}$$
> > This focal length may optionally be scaled by a constant factor to adjust the effective perspective field of view. The principal point $(c_x, c_y)$ is fixed at the image center:
> > $$c_x = \frac{W}{2} \quad \text{and} \quad c_y = \frac{H}{2}$$
> > Finally, we assume zero distortion. The resulting intrinsic matrix ($K$) is thus:
> > $$
> > K =
> > \begin{bmatrix} f_x & 0 & c_x \end{bmatrix} \\
> > \begin{bmatrix} 0 & f_y & c_y \end{bmatrix} \\
> > \begin{bmatrix} 0 & 0 & 1 \end{bmatrix}
> > $$
> > ($K$ is a 3×3 matrix, which cannot be displayed correctly in OpenReview)
> >
> > - **Plücker Embedding:**
> > After obtaining the intrinsic ($K$) and extrinsic ($R, T$) parameters, we utilize the following formula to derive the Plücker Embedding $(T \times d_{u,v}, d_{u,v})$. Here, $d_{u,v}$ denotes the direction vector for the pixel $(u,v)$, which is computed as: $d_{u,v} = R K^{-1} [u, v, 1] + T$. The Plücker Embedding results in a tensor of shape $F \times H \times W \times 6$, which is precisely what is illustrated in Figure 2.
> >
> >
> > [1]CameraCtrl: Enabling Camera Control for Text-to-Video Generation

---

> > > ### Author Response · Authors · 2025-11-21
> > >
> > > **Q3.Novelty Comparison with Panfusion and Text2Light**
> > >
> > > Although both works mention the concept of the sphere, their application scenarios and usage patterns are completely different.
> > >
> > > **(1)Directly transferring the approaches of Panfusion and Text2Light to our task does not work.** Panfusion introduces a two-branch equirectangular–perspective projection attention mechanism. Its core idea is to crop perspective images on the sphere to enhance regional interactions. In practice, they crop 20 perspective images. If this is extended to video generation, it would require 49 frames * 20 images, which cannot fit into GPU memory, making it infeasible. Therefore, compared with Panfusion’s approach, video generation requires a more computation-friendly design. Text2Light proposes spherical continuous representations for interpolation to achieve super-resolution, but this approach cannot be used for a trajectory-controllable panoramic video task. Both works adopt spherical positional encoding. Since the model weights are tightly coupled with the positional encoding, full-model fine-tuning is reasonable in image generation tasks where the model size is relatively small. However, video generation models are much larger and have been trained on millions or even tens of millions of data samples. Changing the positional encoding would prevent us from reusing the previous pretraining, and the available new data is insufficient to retrain a new positional encoding for video. In summary, the designs of Panfusion and Text2Light cannot be directly applied to our task.
> > >
> > > **(2)We make a dedicated design based on the panoramic video generation task.** Since panoramic video generation, compared with image generation, needs to simultaneously maintain temporal consistency and spatial consistency under spherical projection, we design a sphere-aware spatio-temporal mask. Specifically, for a single frame, Text2Light slices patches of equal size on the sphere, but it ignores the sparsity on the sphere; for example, points near the poles undergo more severe projection distortion. When uniformly dividing rectangular patches and projecting them onto the sphere, the polar regions receive more patches than the equatorial regions. Therefore, we exploit this property and specially design our computation scheme to better preserve the characteristics of spherical projection. We compute the great-circle distance, which causes patches near the poles to have more reference patches under the same distance threshold. This enables us to better maintain spherical projection properties.
> > >
> > > Along the temporal dimension, the camera itself is moving; therefore, according to Eq.(4), we compute the rotation of the sphere with respect to the rotation at time 0,
> > >  and dynamically adjust the way we compute the distance between two points $p_1^{t_2}$ and $p_2^{t_2}$ across time. Then, this temporal mask is used as a single shared mask for 3D self-attention, and unlike Panfusion, it does not require additional blocks. This design is tailored to the panoramic video generation task and has not been adopted in previous works on panoramic images.
> > >
> > > ***

---

> > > > ### Author Response · Authors · 2025-11-21
> > > >
> > > > **Q4.Question about High Resolution**
> > > >
> > > >
> > > > We appreciate the reviewer’s suggestion to explore panoramic video super-resolution (SR). However, **we would like to clarify that panoramic video generation and panoramic video SR, while related, but are technically orthogonal problems.**
> > > >
> > > >
> > > >  Our work focuses on controllable panoramic video generation, where the main challenges lie in (i) collecting paired panoramic datasets and (ii) designing specialized architectures for high-quality panoramic synthesis under controllable conditions.**In contrast, panoramic video SR assumes an existing low-resolution panoramic video as input and can be seen as a subsequent refinement step following the generation task.** This leads to a fundamentally different problem formulation and requires its own supervision regime, dataset design, and model architecture. SR focuses on restoring high-frequency details from paired LR–HR panoramic videos, relying on pixel-level supervision and upsampling-oriented modules. Conversely, panoramic video generation must synthesize the entire panoramic sequence from scratch (e.g., from text or conditioning views), emphasizing global scene consistency, temporal coherence, and 360° spatial continuity rather than detail enhancement.A detailed study of panoramic video SR  is therefore beyond the scope of this paper and would constitute an independent research direction.
> > > >
> > > >
> > > > Moreover, existing state-of-the-art panoramic video generation methods such as 360DVD (512×1024) and GenEx (576×1024), as well as state-of-the-art base video generation models (e.g., Wan 2.1 and HunyuanVideo, which at most support 720p outputs), also do not address super-resolution generation. Compared with these works, **our contribution is to demonstrate a feasible framework for controllable, high-quality panoramic video generation,** which can serve as a solid foundation and upstream module for subsequent panoramic video SR and other panoramic video tasks. We will clarify this distinction and the scope of our work in the revised version, and we will treat panoramic video SR as an interesting direction for future work.
> > > >
> > > > ***
> > > >
> > > >
> > > > **Q5. Visualization of trajectories and videos**
> > > >
> > > > Thanks for your suggestion. The side-by-side visualization is indeed important, and we have added it to the new PDF version on page 10. We compare the camera trajectories of our generated video results with the ground-truth cameras, showing strong consistency. Please feel free to reach out if you have any further questions.
> > > >
> > > > ***
> > > >
> > > >
> > > > **Q6.Native Resolution of Datasets**
> > > > Thank you for your interest in our dataset. The native resolution of our data is 2K × 4K. When configuring the rendering resolution in UE, one needs to take into account GPU memory, storage cost, and the requirements of downstream applications. We hope these details are helpful for future research.
> > > >
> > > > ***
> > > >
> > > >
> > > > **Q7. Inferring Exploration Paths from Text Instructions**
> > > >
> > > > Thank you for this insightful question. In the current work, we explicitly assume that the exploration path (camera trajectory) is provided as a controllable input to the model. For benchmark datasets, we use their corresponding ground-truth trajectories, and for in-the-wild cases, we adopt a small set of predefined, commonly used trajectories. Our goal is to study trajectory-controllable panoramic video generation under given trajectories, rather than to infer trajectories from text. Inferring an exploration path purely from text instructions corresponds to a high-level text-to-trajectory planning problem. Recent works such as ChatCam (NeurIPS 2024) and Director3D (NeurIPS 2024) specifically focus on this task and can be viewed as upstream modules that predict trajectories from text. In future work, we plan to explore how to effectively and elegantly integrate such text-to-trajectory planning with our panoramic video generation framework.
> > > >
> > > >
> > > > [1]ChatCam: Empowering Camera Control through Conversational AI
> > > > [2]Director3D: Real-world Camera Trajectory and 3D Scene Generation from Text

---

> > > > > ### Author Response · Authors · 2025-11-21
> > > > >
> > > > > **Q8.Out-of-Distribution Image Performance**
> > > > > We thank the reviewer for this insightful question regarding generalization and dynamic scenes. We respond in three aspects:
> > > > >
> > > > > **(1) Generalization to static real-world scenes.** Although PanoWorld-X is trained only on synthetic data, it exhibits strong zero-shot generalization to out-of-distribution real-world panoramic images. The corresponding results are shown in Supplementary Video 1 – in-the-wild-results.mp4. We test a wide range of styles, including real-captured panoramas, AIGC-generated panoramas, indoor and outdoor environments, underwater scenes, and science-fiction–like imaginary scenes, none of which appear in our training set. Thanks to our carefully designed architecture, PanoWorld-X is able to leverage the priors of the pre-trained video generation backbone and transfer the learned trajectory control to these diverse real-world static scenes. These results demonstrate the effectiveness of our method and verify its practical applicability.
> > > > >
> > > > > **(2) Main contribution of the static synthetic dataset pipeline and controllable generation.** We emphasize that our constructed dataset is, to the best of our knowledge, the first large-scale and diverse synthetic panoramic dataset specifically designed for trajectory control. To address the core challenges of controllable panoramic video generation and preserving spherical projection consistency, it is crucial to first disentangle camera trajectory motion from object motion. Existing synthetic and real-world panoramic datasets either lack accurate trajectory annotations or provide limited scene diversity. Our dataset synthesis pipeline and the curated data therefore fill this gap and offer a solid foundation for future research on trajectory-controllable panoramic video generation and other panoramic-related tasks.
> > > > >
> > > > >
> > > > > **(3) Technical bottlenecks of dynamic panoramic data.** Regarding moving objects, we deliberately focus on static scenes because the technology for generating high-fidelity dynamic synthetic panoramas is still immature. Key technical challenges remain unsolved, such as inserting objects into large-scale virtual environments [1,2], rigging them for animation [3,4], driving physically plausible motion [5,6,7], and ensuring that these motions follow high-level semantic requirements [8,9]. Relying on these immature synthetic pipelines would likely introduce significant noise and artifacts into the training data. Another way is real-world capture, which is extremely labor-intensive, difficult to scale, and often infeasible in complex environments such as glaciers or cliffs where filming is unsafe or impossible.  **As a result, our current model is inherently constrained by the lack of a large-scale dynamic panoramic dataset, and we do not claim strong performance on highly dynamic content.** We explicitly discuss this limitation and the corresponding failure cases in the *FAILURE CASES AND LIMITATIONS DISCUSSION* section (Sec.F in supplementary material ). We therefore treat robust handling of moving objects as a natural next step for future work, as synthetic data generation tools and dynamic panoramic resources continue to evolve.
> > > > >
> > > > > [1]NeRF-Insert: 3D Local Editing with Multimodal Control Signals
> > > > >
> > > > > [2]TIP-Editor: An Accurate 3D Editor Following Both Text-Prompts And Image-Prompts
> > > > >
> > > > > [3]MoDA: Modeling Deformable 3D Objects from Casual Videos
> > > > >
> > > > > [4]MagicArticulate: Make Your 3D Models Articulation-Ready
> > > > >
> > > > > [5]Stable Part Diffusion 4D: Multi-View RGB and Kinematic Parts Video Generation
> > > > >
> > > > > [6]Puppeteer: Rig and Animate Your 3D Models
> > > > >
> > > > > [7]Relightable and Animatable Neural Avatar from Sparse-View Video
> > > > >
> > > > > [8]FantasyHSI: Video-Generation-Centric 4D Human Synthesis In Any Scene through A Graph-based Multi-Agent Framework
> > > > >
> > > > > [9]InteractMove: Text-Controlled Human-Object Interaction Generation in 3D Scenes with Movable Objects

---

### Official Review · Reviewer_4tKR · 2025-11-03

**Soundness:** 4
**Presentation:** 3
**Contribution:** 3
**Rating:** 6
**Confidence:** 4

**Summary:**

The paper proposes PanoWorld-X, a framework for generating high-quality, controllable 360-degree panoramic videos to address the narrow field-of-view and limited user control in existing models. The authors first created a large-scale synthetic dataset of panoramic videos with corresponding 3D trajectories using Unreal Engine. The core of the method is a novel Sphere-Aware Diffusion Transformer, an architecture specifically designed to handle the spherical geometry of panoramic data, which improves visual consistency. This model allows for precise control over camera movement and, according to the paper's experiments, outperforms previous methods in generation quality and control accuracy.

**Strengths:**

1. The paper proposes PanoWorld-X, a new framework that generates high-fidelity, controllable panoramic videos, effectively overcoming the key limitations of narrow field-of-view and lack of interactivity found in prior models.
2. The authors propose a novel pipeline for creating a large-scale panoramic video dataset with accurate trajectory data using Unreal Engine. This addresses the critical lack of high-quality, annotated data needed to train such models.
3. The paper introduces a sphere-aware architecture that adapts pre-trained diffusion model to the spherical geometry of panoramic data, leading to significant improvements in visual quality and spatiotemporal consistency.

**Weaknesses:**

1. The PanoWorld-X framework is fundamentally restricted to generating short video clips due to its reliance on the CogVideoX-5B backbone. This curtails the potential for creating truly long-form, explorable experiences.
2. The model struggles to generate dynamic content, particularly human figures, because its training dataset is composed of static scenes. This limits its ability to create lively and realistic virtual worlds.

**Questions:**

1. The current trajectory sampling method may create a bias towards simple, linear paths in certain environments. Have the authors analyzed the distribution of trajectory complexity, and have more naturalistic sampling strategies been considered to ensure diversity?
2. The Sphere-Aware Attention relies on a fixed, binary distance threshold. Could the authors elaborate on the design rationale for this specific choice? I wonder if the authors have compared their design against other potential strategies, such as using a soft, distance-weighted mask.
3. The introduction of parallel attention branches likely increases computational load. Could the authors provide a quantitative analysis of this overhead?

---

> ### Author Response · Authors · 2025-11-21
>
> Dear Reviewer 4tKR, we sincerely appreciate your recognition of our paper's contributions, particularly the novel framework for controllable panoramic video generation, the valuable dataset we introduced, and the effective quality enhancement method we developed. Your acknowledgment underscores the significance of our work in advancing this research field.We would like to take this opportunity to respond to your comments.
>
> **Q1. Questions about Long-Video Generation**
>
> Current research faces two largely independent challenges: Current research faces two largely independent challenges: (1) video generation base models still struggle to produce long video sequences, and (2) existing base models cannot generate high-quality panoramic videos. Long-video generation research primarily focuses on issues such as information compression, memory mechanisms, memory costs, etc. In contrast, panoramic video generation mainly concentrates on how to construct suitable datasets, preserve the properties of spherical projection, effectively leverage prior knowledge, etc. **Long-video generation and panoramic-video generation correspond to distinct problem formulations, requiring different supervision regimes, dataset constructions, and architectural designs.** Our work primarily addresses the second problem—adapting video base models for high-fidelity panoramic video generation. Compared with prior panoramic video generation approaches, which also do not support long videos, we provide a large paired dataset and a sphere-aware architecture that more effectively leverages pretrained video diffusion models. Progress in long-video generation research will naturally benefit panoramic long-video synthesis as well. However, without dedicated efforts on how to convert standard video models into capable panoramic generators, this area would remain largely unexplored.
>
> **Furthermore, although our model generates only 49 frames, it still enables long-range spatial exploration, allowing the viewer to observe distant small objects.** For example, in supplementary video 1-in-the-wild-results.mp4, the first case (0–3 s) shows the camera moving all the way toward the far-off mountains, causing the boats in the foreground to disappear. In the fourth case (9–11 s), the camera moves toward objects on a distant shelf, even though these items are barely visible in the first frame. Thanks to the model’s ability to support large-magnitude movements, it can explore scenes extensively even though the base model itself does not generate long video sequences.
>
> ***
>
> **Q2. Question about Dynamic Content Generation**
>
> We thank the reviewer for this insightful question regarding dynamic scenes. We respond in two aspects:
>
> **(1) Technical bottlenecks of dynamic panoramic data.** The temporal stability of non-rigid objects depends not only on network design but also on the construction of appropriate datasets. We deliberately focus on static scenes because the technology for generating high-fidelity dynamic synthetic panoramas is still immature. Key technical challenges remain unsolved, such as inserting objects into large-scale virtual environments [1,2], rigging them for animation [3,4], driving physically plausible motion [5,6,7], and ensuring that these motions follow high-level semantic requirements [8,9]. Relying on these immature synthetic pipelines would likely introduce significant noise and artifacts into the training data. Another way is real-world capture, which is extremely labor-intensive, difficult to scale, and often infeasible in complex environments such as glaciers or cliffs where filming is unsafe or impossible.  **As a result, our current model is inherently constrained by the lack of a large-scale dynamic panoramic dataset, and we do not claim strong performance on highly dynamic content.** We explicitly discuss this limitation and the corresponding failure cases (Sec.F in supplementary material ). We therefore treat robust handling of moving objects as a natural next step for future work, as synthetic data generation tools and dynamic panoramic resources continue to evolve.
>
> **(2) Main contribution of the static synthetic dataset pipeline and controllable generation.** We emphasize that our constructed dataset is, to the best of our knowledge, the first large-scale and diverse synthetic panoramic dataset specifically designed for trajectory control. To address the core challenges of controllable panoramic video generation and preserving spherical projection consistency, it is crucial to first disentangle camera trajectory motion from object motion. Existing synthetic and real-world panoramic datasets either lack accurate trajectory annotations or provide limited scene diversity. Our dataset synthesis pipeline and the curated data therefore fill this gap and offer a solid foundation for future research on trajectory-controllable panoramic video generation and other panoramic-related tasks.

---

> > ### Author Response · Authors · 2025-11-21
> >
> > **Q3. Distribution of trajectory complexity**
> >
> > Thank you for your attention to our dataset. We have analyzed the distribution of trajectory complexity in Supplementary B.3(which can be found after the references in the main paper, or in appendix.pdf in the submitted zip). We use tortuosity (defined as the ratio between the  straight-line distance and the  actual path length between the start and end points) to quantify trajectory complexity. Our analysis shows that only a small portion of trajectories are nearly perfectly straight; the majority contain at least one or two noticeable directional changes, and the remaining trajectories exhibit rapid turns. This distribution indicates that our trajectories are reasonably diverse rather than being dominated by simple linear paths. We reproduce the supplementary results below for reference.
> >
> > | Tortuosity Range     |  Percentage   | Meaning Interpretation  |
> > |----------------|---------|----------------|
> > | 1                   | 7.13%   | Perfectly straight path    |
> > | 1>T≥0.75      | 60.24%   | Noticeable directional changes, possibly with one or two turns      |
> > | 0.75>T ≥0.5     | 32.29%   | Significant directional changes, possibly with multiple turns       |
> > | 0.5>T ≥0     | 0.34%   | Highly complex trajectories          |
> >
> > Regarding  naturalistic sampling strategies, we briefly explain our design in Lines 213–218, and elaborate here. Our sampling strategy is explicitly inspired by how a human would move in a scene. First, a person needs to understand where obstacles are and avoid walking through them; correspondingly, we extract mesh surfaces and perform collision checking to identify non-traversable regions. Second, a person plans a path from a starting position to a destination; analogously, we randomly sample start and end points and then use Dijkstra’s algorithm to compute the shortest path between them. Due to the presence of obstacles, these shortest paths naturally contain turns rather than being straight lines. Finally, to mimic the smoothness of human motion, we apply Laplacian smoothing to reduce abrupt changes in the trajectory.
> >
> > Overall, this pipeline yields trajectories that are both diverse in complexity and naturalistic in the sense of human-like navigation behavior, rather than artificially simple or overly linear paths.

---

> > > ### Author Response · Authors · 2025-11-21
> > >
> > > **Q4. Attention mask setting**
> > >
> > > Thanks for pointing out the unclear description of the attention mask setting. Our attention mask is set as a binary (True/False) mask rather than a soft mask. The reasons are as follows:
> > >
> > > (1) We follow the implementation of the original torch.nn.functional.scaled_dot_product_attention. Since it is applied to causal attention, it only supports boolean (True/False) values to indicate activation. To preserve the pre-trained weights as much as possible, we maintain consistency with the original attention configuration.
> > >
> > > (2) If we were to adopt a soft distance-weighted approach, the mask would be added to the attention weights ($Q \times K$). However, since the value range of attention weights varies across different layers and models, it is difficult to determine the appropriate maximum value for distance decay. Arbitrary settings could disrupt the original weight information and lead to training instability.
> > >
> > >
> > > ```python
> > > def scaled_dot_product_attention(query, key, value, attn_mask=None):
> > > ……
> > > attn_bias = torch.zeros(L, S, dtype=query.dtype, device=query.device)
> > > if is_causal:
> > >         assert attn_mask is None
> > >         temp_mask = torch.ones(L, S, dtype=torch.bool).tril(diagonal=0)
> > >         attn_bias.masked_fill_(temp_mask.logical_not(), float("-inf"))
> > > if attn_mask is not None:
> > >         iif attn_mask.dtype == torch.bool:
> > > attn_bias.masked_fill_(attn_mask.logical_not(), float("-inf"))
> > >         else:
> > >  attn_bias = attn_mask + attn_bias
> > > attn_weight = query @ key.transpose(-2, -1) * scale_factor
> > > attn_weight += attn_bias
> > > attn_weight = torch.softmax(attn_weight, dim=-1)
> > > ……
> > > ```
> > >
> > >
> > > (3) Given the aforementioned limitations, while we did not implement a soft distance-weighted strategy, we investigated the impact of the spatial range by conducting an ablation study on the threshold in Eq. (5) (which determines the maximum distance for two points to be considered in sphere attention). This experiment is detailed in Supplementary Table E and summarized below.
> > > When the model degenerates to pure self-attention (threshold = 1), the increased trainable parameters result in only marginal performance improvements. However, due to the lack of explicit geometric priors, the model struggles to achieve higher performance levels.
> > > Conversely, reducing the threshold shrinks the receptive field, weakening the model’s capacity to capture global panoramic distortions.
> > > Although reducing the threshold yields lower performance compared to the full model, it still surpasses the baseline (without Sphere-Aware Attention).
> > > This demonstrates that the setting of the mask distance directly influences the degree to which spherical geometric priors are incorporated. An appropriate distance setting provides beneficial regularization, thereby enhancing model training.
> > >
> > >
> > > | Model      | PSNR    |  SSIM |  LPIPS   | FID |  FVD |$ R_{err}$ | $T_{err}$ |
> > > |----------------|---------|----------------|----------------|----------------|----------------|----------------|----------------|
> > > | Baseline(w/o Sphere-Aware Attention) | 17.59   |     0.56     | 0.27   |     29.96     |492.98   |     0.069  |  0.076   |
> > > | Self-Attention(Threshold=1)   | 17.94  |    0.58    | 0.29   |     29.03     | 489.03   |     0.065    |  0.077  |
> > > | Threshold=0.25   | 18.68   |    0.61     | 0.25   |     28.77    | 473.01   |     0.062   | 0.075   |
> > > | Full Model(Threshold=0.5)    | **19.34**   |     **0.63**     | **0.24**   |    **28.01**     |**467.18**   |  **0.061**     |  **0.073**   |
> > >
> > > ***
> > > **Q5. Computational Cost Analysis**
> > >
> > > Indeed, our additional branch increases both memory usage and computation, which is essentially a trade-off between performance and cost. We conducted a quantitative evaluation: the original CogVideoX-5B-I2V requires 5G of memory for inference and about 200s on a single A800 GPU to generate a 49-frame video, while our model requires 8G of memory and about 280s.The original model LoRA fine-tuning model is 63 GB, while our model is 78 GB. We believe this trade-off is worthwhile, because it not only improves visual detail and quantitative metrics, but also enhances the consistency of spherical projection on the panoramic sphere, especially the left-right consistency (as shown in Supplementary Video4-ablation-sphere-attn.mp4). This is crucial for panoramic applications, since on head-mounted displays or when converting to a perspective view, visible vertical seams at the left and right boundaries can significantly degrade the immersive experience.

---

> > > > ### Author Response · Authors · 2025-11-21
> > > >
> > > > [1]NeRF-Insert: 3D Local Editing with Multimodal Control Signals
> > > >
> > > > [2]TIP-Editor: An Accurate 3D Editor Following Both Text-Prompts And Image-Prompts
> > > >
> > > > [3]MoDA: Modeling Deformable 3D Objects from Casual Videos
> > > >
> > > > [4]MagicArticulate: Make Your 3D Models Articulation-Ready
> > > >
> > > > [5]Stable Part Diffusion 4D: Multi-View RGB and Kinematic Parts Video Generation
> > > >
> > > > [6]Puppeteer: Rig and Animate Your 3D Models
> > > >
> > > > [7]Relightable and Animatable Neural Avatar from Sparse-View Video
> > > >
> > > > [8]FantasyHSI: Video-Generation-Centric 4D Human Synthesis In Any Scene through A Graph-based Multi-Agent Framework
> > > >
> > > > [9]InteractMove: Text-Controlled Human-Object Interaction Generation in 3D Scenes with Movable Objects

---

### Author Response · Authors · 2025-12-03
**General Response**

We would like to sincerely thank all the reviewers and the area chair for their time in reviewing our paper and for their valuable suggestions.

***

**Our Contributions and Strengths**

**1.Novel Framework:** We introduce a new framework for **generating high-quality, trajectory-controllable panoramic videos.** Compared with previous view-controllable video generation models, our method overcomes the narrow field-of-view limitation, resulting in a much stronger sense of immersion. Compared with existing panoramic video generation models, our approach further enables trajectory control, allowing users to freely explore the scene, while also achieving higher visual quality. The novelty and advantages of our task formulation have been recognized by Reviewers 4tKR and 8Add.

**2.New Dataset:** To achieve the pipeline, we design **a dataset curation pipeline based on Unreal Engine and introduce the PanoExplorer dataset, which contains 110K panoramic videos with trajectory annotations.** Reviewer kKFp considers our data curation pipeline well designed. Reviewer 4tKR commends that we address the lack of high-quality annotated data in this domain, and Reviewers 3MJN and 8Add appreciate that our dataset is valuable for the field and for future research.

**3.Well-motivated module:** To bridge the gap between pretrained diffusion models and the spherical geometry of panoramic data, we introduce a **sphere-aware DiT block that leverages spherical connectivity to enhance spatiotemporal consistency.** This module is acknowledged by Reviewers 8Add, kKFp, and 3MJN as being theoretically well-motivated because it explicitly models the spherical topology. Reviewer 4tKR further notes that this module brings significant improvements in both visual quality and spatiotemporal consistency.

**4.Performance:** Our extensive comparisons and ablation studies demonstrate clear improvements over the baselines, which is acknowledged by Reviewers 3MJN and kKFp.

***

**Main Questions and Corresponding Responses**

Before November 27 (OpenReview information leak announcement), only one reviewer had responded to our rebuttal.

**Reviewer kKFp stated that all of their concerns had been addressed and  `raised the score from 6 to 8` on November 26, 18:15 (AoE).**


***
**For Reviewer 4tKR**, who initially gave our paper a **positive score of 6**, we summarize our responses as follows:

| Question | Response | Index |
|---|---|---|
| Support for more features | Clarify that this is beyond the scope of our paper | Q1,Q2 |
| Trajectory complexity | Add detailed trajectory analysis and trajectory sampling strategies | Q3 |
| Attention details | Add code explanations and ablation studies | Q4 |
| Computational details | Add training and inference GPU memory and time costs | Q5 |

***

**For Reviewer 8Add**, although the reviewer did not respond to the rebuttal, **the initial review explicitly indicated a willingness to adjust the score if these concerns were satisfactorily resolved.** We summarize our responses as follows:

| Question | Response | Index |
|---|---|---|
| Architectural contribution | Explain that directly transferring previous methods fails; highlight our unique design for the task | Q1, Q3 |
| Route representation details | Add detailed formula explanations | Q2 |
| Super-resolution | Clarify that this is a downstream application beyond the scope of our paper | Q4, Q6 |
| Visualization | Add side-by-side visualizations in the updated PDF | Q5 |
| Out-of-distribution performance | Due to current bottlenecks in dynamic panoramic video synthesis, this issue lies beyond the scope of this work; focus on static panoramic dataset synthesis and controllable static generation | Q8 |

***

**For Reviewer 3MJN**, we summarize our responses as follows:

| Question | Response | Index |
|---|---|---|
| Synthetic vs. real-world differences | Explain Unreal Engine's capabilities and limitations; state that ours is currently the highest quality synthetic dataset | Q1 |
| Dynamic cases discussion | Due to current bottlenecks in dynamic panoramic video synthesis, this issue lies beyond the scope of this work; focus on static panoramic dataset synthesis and controllable static generation | Q2 |
| Computational details | Add training and inference GPU memory usage and time costs | Q3 |
| Hard case performance | Analyze performance on three complex cases to demonstrate capability boundaries | Q4 |
| Exploration-aware attention ablation | Add two new ablation studies | Q5 |

---

> ### Author Response · Authors · 2025-12-03
>
> To clearly illustrate the problems we address in this work, we summarize them as follows.
>
> **1. Proposed Dataset – More Details.** Reviewer 4tKR raised a concern about trajectory complexity, and Reviewer kKFp asked about the trajectory distribution. In response, we added a trajectory tortuosity statistic to demonstrate that our dataset exhibits diverse trajectory patterns (see Q3 for Reviewer 4tKR). We also analyzed the curvature of trajectories and categorized them into six typical types, providing detailed statistical analysis (see Q1 for Reviewer kKFp).
>
> **2.Method Details.** Reviewer 8Add requested more details about the Exploration Route Representation, while Reviewer kKFp asked about the meaning of the dataset angles and the details of scale normalization. We have added more detailed mathematical formulations and theoretical explanations addressing these points: see Q4 and Q5 in our response to Reviewer kKFp, and Q2 in our response to Reviewer 8Add.
>
> **3.More Evaluation.** Reviewer kKFp raised concerns about the generalization ability to real videos and about the performance of baseline models after fine-tuning on our dataset. In response, we added experiments for both settings, as reported in Q1 and Q3 in our response to Reviewer kKFp. Before the information leak was flagged on OpenReview, Reviewer kKFp stated that these additions addressed their concerns and increased the score.
>
> **4.Computational Cost.** We provide detailed analysis results on inference speed and memory consumption (see Q5 in our response to Reviewer 4tKR and Q3 in our response to Reviewer 3MJN). We demonstrate that the slight increase in computational cost is a worthwhile trade-off, as it not only improves visual details but also leads to better quantitative metrics.
>
> **5.Support for additional features.** Reviewers kKFp and 8Add requested support for super-resolution and longer videos .**We clarify that the requests for super-resolution  and longer videos  are beyond the scope of this paper.** Our work focuses on controllable and high-quality panoramic video generation, while long-duration video generation and super-resolution are downstream applications. These tasks target different core problems and therefore require their own dataset curation, model architecture, and supervision signals. We discuss these points in Q1 of our response to Reviewer 4tKR and Q4 of our response to Reviewer 8Add.
>
> **6.Dynamic Content Generation Ability.** Reviewers 4tKR, 8Add, and 3MJN all inquired about the performance on dynamic content. **We would like to clarify that this question is fundamentally constrained by the limited availability and construction of dynamic panoramic datasets and lies beyond the scope of our paper.** At present, there are significant technical bottlenecks in collecting large-scale dynamic panoramic data; even large-scale static trajectory-paired panoramic video datasets did not exist prior to our work. Our main contribution is therefore to first address this challenge by constructing such a large-scale static panoramic video dataset and achieving high-quality static panoramic video generation.

---

### Meta-Review · Area_Chair_kL4J · 2025-12-26

**Summary:**

This paper proposes PanoWorld-X for generating high-quality, controllable 360-degree panoramic videos to address narrow filed-of-view and insufficient controllability challenges. Furthermore, a novel panorama dataset termed PanoExplorer containing over 11K panoramic videos paired with trajectories by Unreal Engine is provided.

Reviewers agree that the proposed sphere-aware architecture is theoretically well motivated, and effectively addresses the geometric misalignment between panoramic data and standard diffusion priors. The PanoExplorer dataset containing large-scale and accurately labeled video-trajectory pairs is a valuable contribution.

Reviewers’ main concerns are:
1) The PanoExplorer is solely comprised of synthetic data, which may exhibit significant domain gap with real-world panoramic data. No further details of resolutions or trajectory distributions are provided. (8Add, 3MJN, kKFp)
2) PanoWorld-X framework is restricted to generating short video clips with limited output resolution (480*960). (4tKR, 8Add)
3) The model also struggles to generate dynamic content (e.g., human figures, weather changes), which limits its ability to create realistic virtual worlds. (8Add, 4tKR, 3MJN)
4) The novelty of the architectural contribution is limited. Trajectory mechanism like ControlNet and the core mechanism, sphere-aware attention, are discussed in previous panoramic image generation. (8Add)
5) Current trajectory sampling method may create bias towards simple paths. Further complexity analysis needs to be provided. (4tKR, 3MJN)
6) Some technical details need to be clarified and extended. (4tKR, 8Add, kKFp)
7) Evaluation results need to be visualized. Evaluation on datasets from other sources is also needed to ensure comparison fairness. (8Add, kKFp)
8) The computation load introduced by parallel attention branches needs more analysis. (4tKR, 3MJN)

In summary, this paper was reviewed by four experts in the field. The recommendations are 6, 4, 6, 4. The reviewers raised multiple concerns as listed above.  However, after rebuttal, the proposed PanoExplorer framework still demonstrates limitations regarding novelty. And concerns on the scope of generation (resolution and dynamic videos) and the domain gap in the dataset remain unsolved.

**Reviewer Concerns:**

**Well addressed:**
- Analyses on trajectory sampling method is provided.
- Multiple technical details are specified.
- Computational overhead analysis is provided.

**Partly addressed:**
- PanoExplorer provides the first large-scale panoramic video synthesis framework. Although domain gap between synthetic data and real-world data exists, simulation is approximated with real data as much as possible. More analyses on datasets are provided.
- Instead using datasets from another source, other methods are retrained on the proposed dataset, whose overall performance does not surpass the proposed model.

**Unsolved:**
- High-resolution and highly dynamic video clip generation is a challenging topic beyond this work.

**Reviewer Scores:**

**3MJN (4)**

Most concerns raised by 3MJN are resolved. The reviewer might raise the score.

**kKFp (6)**

Most concerns raised by kKFp are resolved. The reviewer raised the score to 8.

**4tKR (6)**

Concern 2, 3 and 4 raised by reviewer 4tKR remain unsolved. 4tKR may keep the score.

**8Add (4)**

Concern 2, 3 and 4 raised by reviewer 8Add remain unsolved. 8Add may keep the score.

---

### Decision · Program_Chairs · 2026-01-26

Reject